

# Effects of nitrogen doses on stomatal characteristics, chlorophyll content, and agronomic traits in wheat (*Triticum aestivum* L.)

Fatih Oner

Field Crops/Agricultural Faculty, Ordu University, Ordu, Turkey

## ABSTRACT

It is very important to determine the chlorophyll content (SPAD) and nitrogen (N) requirement in order to increase the seed yield and nutritional quality of wheat. This research was carried out with three N doses (0, 50, 100 kg ha$^{-1}$) and nine wheat cultivars (Alpu-2001, Soyer-02, Kate-A1, Bezostaja-1, Altay-2000, Müfitbey, Nacibey, Harmankaya-99 and Sönmez-2001) during 2-years field condition according to factorial randomized complete block design and three replications. In this study, with the increase of N dose (N50), seed yield increased by 13%, plant height by 10.8%, 1,000 seed weight by 10.5% compared to control plants (N0). The increase of N dose from 50 kg ha$^{-1}$ to 100 kg gave lower increase rates in the same criteria (11.7%, 11.4%, 10.3%, respectively). However, the spike number per plant, spikelet number in spike, seed number in spike, spike length showed statistically significant differences between N doses and varieties. Boost of N doses caused a significant increase compared to plants without N application. The chlorophyll content and flag leaf area index were determined at three growth times (1$^{st}$ growth time; early, 2$^{nd}$ growth time; the middle and end of flowering, 3$^{rd}$ growth time; with a 10-day interval). Chlorophyll content was significantly ($p < 0.01$) affected by the N dose, variety and growth time. As N doses increased, chlorophyll content increased, and it was higher at both N doses compared with N0. The chlorophyll content had the highest rates (30.22%) at 1$^{st}$ growth time and it decreased as the growth period progressed. N doses, varieties and their interactions had significant effects on the flag leaf area index. The highest flag leaf area index (41.9 cm$^{2}$) was determined from variety Bezostaja-1 and 100 kg ha$^{-1}$ N dose treatment. The effect of N dose was found significantly on abaxial and adaxial stomata width-length and epidermal cells. The adaxial and abaxial stomata width were higher than N0 at both N levels. The highest adaxial and abaxial stomata width- length was obtained from 100 kg ha$^{-1}$ N dose. As nitrogen concentration increased, both stomatal density and stomatal index increased. The stomatal index varied between 19% and 36%. The lowest stomata density had appeared in the 100 kg ha$^{-1}$ N dose and Bezostaja-1 variety. As a result, stomatal characteristics, chlorophyll content, and agronomic traits of wheat were significantly affected by increasing N doses.

Corresponding author
Fatih Oner, fatihoner38@gmail.com

## INTRODUCTION

Cereals and products obtained from seeds are an indispensable source of basic nutrition for the majority in Turkey, as in many countries of the world. Cereals are the most important nutritional sources in human nutrition. More than 50% of the calorie input in nutrition is covered by seeds. Wheat is an extremely useful plant in terms of meeting people's calorie and protein needs. At 20% of the world's calorie and protein consumptions is met by the wheat plant. Wheat (*Triticum* L.) is one of the first cultivated cereal and has been the main food of people in Europe, Western Asia and North Africa for thousands of years. In Turkey, wheat ranks first in terms of both cultivation area and production potential. According to 2020 data, wheat constituted 28% of the seed cultivation area and 65% of seed production while wheat is cultivated in 44% of the total cultivated area in Turkey (*FAOSTAT, 2020*).

The seed demand increases by the population. Both increasing yields for the rising demand for seed production and ensuring sustainability are the biggest challenges in agriculture. Fertilization is required for high efficiency in agricultural systems.

Nitrogen (N) is the most important macronutrient limiting plant growth (*Hawkesford, 2014*; *Yang et al., 2019*). N greatly boosts crops (*Lachutta & Jankowski, 2024*). N fertilization delays leaf aging by reducing N release from the leaf after flowering and provides carbohydrates for seed filling (*Li et al., 2012*). Plant growth and photosynthesis are inhibited in N deficiency, which causes a decrease in yield (*Jin JiYun & He Ping, 1999*; *Shangguan, Shao & Dyckmans, 2000*). Also, vegetative and reproduction are delayed, and yield components such as yield and agronomic traits, the number of spikes per plant, number of seeds per spike, single seed weight and number of seeds per plant decrease (*Wang et al., 2021*). Additionally, N deficiency reduces leaf area index and protein content of the plant and seed (*Steer & Harrigan, 1986*). N deficiency increases flower abscission, reduces seed formation and inhibits seed growth (*Dordas & Sioulas, 2008*; *Ciampitti & Vyn, 2011*). N accumulation and productivity increase are closely related to each other (*Lemaire et al., 2021*). The fact that N uptake by plants affects the increase in productivity shows (*Kayan et al., 2020*) that N physiologically plays a role in the formation of new plants (*Sinclair, Rufty & Lewis, 2019*).

Crop production occurs through photosynthesis and dry matter accumulation depending on leaf area index (*Lee & Tollenaar, 2007*; *Sadras et al., 2016*; *Thomas & Ougham, 2014*). Photosynthetic products constitute the main source of plant biomass. Therefore, as photosynthesis in the leaf increases, so does the biomass obtained (*Noor et al., 2023*). N affects leaf area index development (*Steer & Harrigan, 1986*). Seed N concentration and seed yield are the main targets in wheat breeding, but it has been difficult to increase them simultaneously due to the negative genetic relationship between them (*Mu et al., 2018*). When N is applied at optimum levels in wheat, 35–42% of the N accumulated in the soil is in the leaf lamina, 14–20% in the leaf sheath, 20–31% in the stem and 16–23% in the spike (*Slafer, Andrade & Satorre, 1990*; *Bogard et al., 2010*; *Pask et al., 2012*).

Stomata are natural microscopic pores found in the leaf that serve to connect near part of the leaf to the external environment (*Barraclough et al., 2010*; *Gaju et al., 2014*). Stomata are surrounded by a pair of protective cells and enable gas exchange between the inner part of the leaf and the ex-ternal environment. The photosynthetic capacity of plants depends on stomatal size and density. The stomatal density of the leaf, that is, the number of stomata per unit area of the leaf, is an important parameter affecting gas exchange and is important in terms of the water and carbon relations of the plant (*Hedrich & Shabala, 2018*; *Lawson & Vialetchabrand, 2019*). Stomatal movement is closely related to N. Nitrate is the main source in plant development, is important in regulating stomatal movements (*Rogiers, Hardie & Smith, 2011*). It has been reported by many researchers that high N fertilisation increases photosynthesis and plant growth (*Evans, 1989*; *Doheny-Adams et al., 2012*; *Mu & Chen, 2021*). At 75% of the N in the leaf is found in chloroplasts and is a part of ribulose bisphosphfate carboxylase (Rubisco). Reduced photosynthesis in the presence of low N is generally due to a decrease in chlorophyll content and Rubisco activity (*Huber, Sugiyama & Alberte, 1989*; *Cechin & De Fátima Fumis, 2004*). Since N is a component of chlorophyll (*Robinson & Burkey, 1997*), in some species wheat (*Triticum aestivum* L.), stomatal conductance increases with N (*Evans & Terashima, 1987*).

This study is based on the hypothesis that increasing N doses in wheat influences stomatal structure and chlorophyll content.

## MATERIALS AND METHODS

### Experimental design and materials

The field study was conducted at the experimental farm of Ordu University, Faculty of Agriculture, Turkey (40°58′09″ N 37°56′19″ E) in two wheat growing seasons (2020–2021 and 2021–2022). The experiment was carried out according to the factorial randomized complete block design with 3 repetitions and for 2 years. The soil was neutral in response to (pH 5.78–6.62), somewhat salty (0.034–0.046%), with very low N (0.012–0.016 kg ha$^{-1}$), low phosphorus (0.20–0.19 kg ha$^{-1}$), high potassium (4.07–4.28 kg ha$^{-1}$), and saturation (%) at 2020–2021 and 2021–2022 growing periods, respectively (Table 1). Nine wheat varieties (Alpu-2001, Soyer-02, Kate-A1, Bezostaja-1, Altay-2000, Müfitbey, Nacibey, Harmankaya-99, Sönmez-2001) were used in the study, which were grown varieties in Turkey. In the experiment, 60 kg phosphorus and 50 kg calcium fertilizers were applied to the hectare before sowing and half of them were applied with sowing and the other half before emergence. Three rates of N dose (0 (control), 50 and 100 kg ha$^{-1}$) were applied as granules of urea (34.5% N). The experimental plots were 3 m × 5 m and consisted of 5 rows 0.5 m apart. Border rows were not included for any sampling. Potassium, as potassium sulfate (50% potassium) and phosphorus, as calcium superphosphate (43% phosphorus), were applied for each plot as basal fertilizer at rates.

The meteorological data during the growing seasons of wheat were given in Fig. 1. The region of investigation has a temperate climate, humid summers followed by rainy and warm winters. During the 2020–2021 growing season, the mean humidity ranged between 65.3% and 78.2%, while as in the 2021–2022 growing season, it ranged between 60.9% and

**Table 1 Some chemical characteristics of the soil in the field area.**

| Years | Salt (%) | *Saturation (%) | pH | N (kg ha$^{-1}$) | P (kg ha$^{-1}$) | K (kg ha$^{-1}$) |
|---|---|---|---|---|---|---|
| **2021** | 0.034 | 118.8 | 5.78 | 0.012 | 0.20 | 4.07 |
| **2022** | 0.046 | 116.6 | 6.62 | 0.016 | 0.19 | 4.28 |

**Note:**
  * Saturation > 110% is highly-clay.

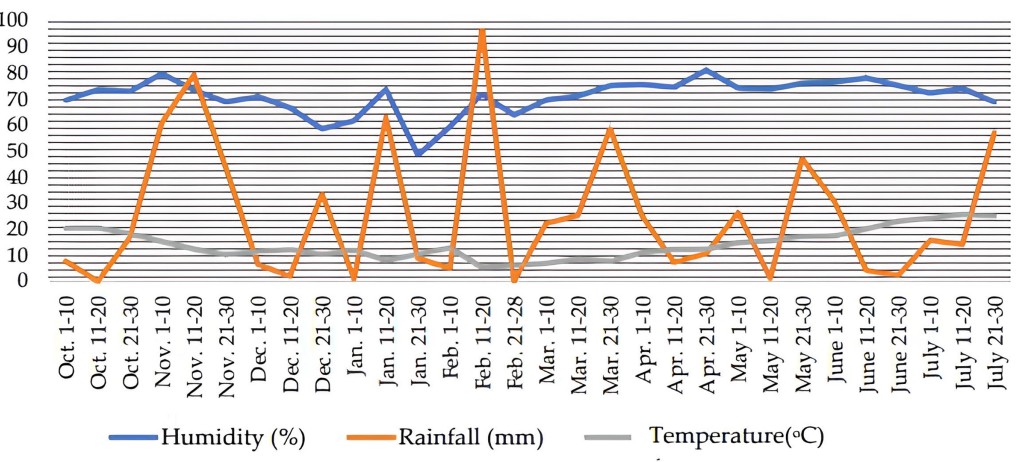

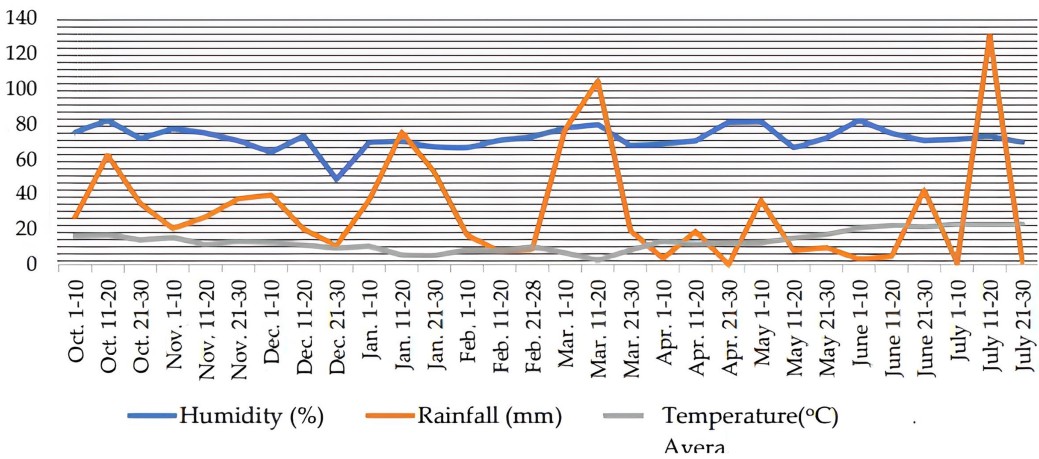

**Figure 1 Climatic data for 10 days each during the growing season (2020–2021 and 2021–2022) from sowing to harvest at Ordu, Turkey.**

77.1%. Average precipitation and temperature values obtained in all years were close to each other (Fig. 1).

# METHODS OF DETERMINATION

## Yield and agronomic traits

Plants were harvested at the maturity time, 270 days after planting. Ten plants in each pod were selected randomly to measure yield components, as seed yield, plant height, 1,000

seed weight, the number of spikelets per spike, the number of seeds per spike, spike length, number of spikes. All observations and measurements were made according to the International Union for the Protection of New Varieties of Plants (UPOV) criteria.

### Chlorophyll content and flag leaf area index

The chlorophyll content (SPAD value) and flag leaf area index were determined on the flag leaves of selected ten plants of every plot in three growth times (1st growth time; early, 2nd growth time; the middle and end of flowering, 3rd growth time; with a 10 day interval). Chlorophyll content (SPAD value) was measured by the Minolta SPAD-502 chlorophyll meter (Minolta, Tokyo, Japan). The measurements were done five times for each flag leaf, and the mean was calculated as the SPAD value of the given flag leaf. Leaf area measurements were made using a LI-3000 area meter (Li-Cor Inc., Lincoln, NE, USA). Both chlorophyll content and leaf area measurements were measured from the same flag leaves of the same ten selected plants.

### Characteristics of stomata

Stoma measurements were made on the flag leaves of the plants during the antesis period. Epidermis of the leaves were cleaned with ethanol. The transparent nail polish was applied on the abaxial and adaxial sides of the flag leaves to form a thin layer. The nail polish was allowed to dry for 20–30 min. The nail polish layer was peeled off using tweezers and placed on a glass slide for observation under a microscope. Parameters like stomatal density (no/mm$^2$), stoma length (μm), stoma width (μm), stomatal index (%) and epidermal cells (no/mm$^2$) were recorded. Three field of view during observation were randomly selected for each slide and the number of stomata in each field of view was counted. The stomatal density was calculated on a leaf surface basis, and stomatal index was calculated as: number of stomata/(number of stomata + number of epidermal cells) (Zhao et al., 2005).

## STATISTICAL ANALYSIS

In order to decide on the analysis method to be performed, Skewness and Kurtosis values were examined (±3.29) and it was determined that the data were compatible with normal distribution (±1.36). Factorial randomized complete block design (RCBD) (Model 1) was used in the study and variety and N dose were considered as fixed factors. Comparisons between years were determined by t test.

$$\gamma_{ijk} = \mu + \alpha_i + \beta_j + (\alpha\beta)_{ij} + \varepsilon_{ijk} \tag{1}$$

$\alpha$: Variety
$\beta$: N doses
$\alpha\beta$: Variety X N dose
$\varepsilon$: Error

In order to determine the appropriate multiple comparison test, the Levene homogeneity test was performed and when it was determined that the data were homogeneous, the TUKEY multiple comparison analysis was decided. The TUKEY

(Model 2) multiple comparison test was used at the 0.01 or 0.05 level, depending on the degree of significance, if there was a significant difference between the components.

$$HSD = qx\sqrt{\frac{MSW}{n}} \tag{2}$$

Analysis of variance was performed with SPSS 24.0 software (SPSS Inc., IL, USA).

In this study, principal component analysis (PCA) was applied to enable better analysis of the high-dimensional dataset and to optimize the impact of dimensions during the modelling process. PCA evaluated the relationships among the variables in the dataset and identified a few principal components that explain the variance best.

PCA was conducted using the "FactoMineR" package (Lê, Josse & Husson, 2008) and visualized with "factoextra" (Kassambara & Mundt, 2020) in R. The dataset was standardized to ensure comparability across variables. The N dose groups (N0: 0 kg ha$^{-1}$, N1: 50 kg ha$^{-1}$, N2: 100 kg ha$^{-1}$) and wheat varieties (V1–V9) were included as a grouping factor separately, and confidence ellipses were added to illustrate variability within each group.

Variable contributions to the first two dimensions (Dim1 and Dim2) were analyzed to identify key traits driving the observed patterns. Results were visualized with a biplot combining individuals and variables, and a variable contribution plot was created to highlight the relative influence of each variable.

## RESULTS

All parameters except plant height and spike length values were not statistically affected by the years and no significant difference was observed between 2020–2021 growing season and 2021–2022 growing season (Figs. 2A, 2B).

### Seed yield and agronomic traits

The effects of variety and N dose on yield and agronomic traits of wheat plant are given in Tables 2 and 3 both years. With the increase in nitrogen dose, an increase was observed in all parameters examined compared to the control and N2 dose gave the highest values. When the N dose increased from 0 to 100 kg ha$^{-1}$, the change in seed yield increased from 3,659 to 5,511 kg ha$^{-1}$ (increased 50.61%) at first year (Table 2), while from 3,859 to 5,696 kg ha$^{-1}$ rise was obtained (increased 47.60%) at second year (Table 3). The highest seed yields were obtained from Müfitbey (5,044 kg ha$^{-1}$ for 2020–2021 and 5,222 kg ha$^{-1}$ for 2021–2022 growing seasons respectively) and Kate-A1 (4,900 kg ha$^{-1}$ for 2020–2021 and 5,067 kg ha$^{-1}$ for 2021–2022 growing seasons, respectively) varieties (Tables 2 and 3).

The plant height was significantly affected by N dose and varieties, but it wasn't significantly affected by their interactions. As the N dose increased from 0 to 50 kg ha$^{-1}$, the plant height increased from 72.25 to 85.24 cm (increased 17.97%). Morever its increase from 50 to 100 kg ha$^{-1}$ resulted in the the plant height growth from 85.24 to 96.94 cm (increased 28.23%) at first year (Table 2). In the second year, these increase rates were measured as 1.00% and 15.26%, respectively (Table 3). While the lowest plant height mean was obtained from Müfitbey wheat variety (76.70 cm for 2020–2021 and 81.34 cm for

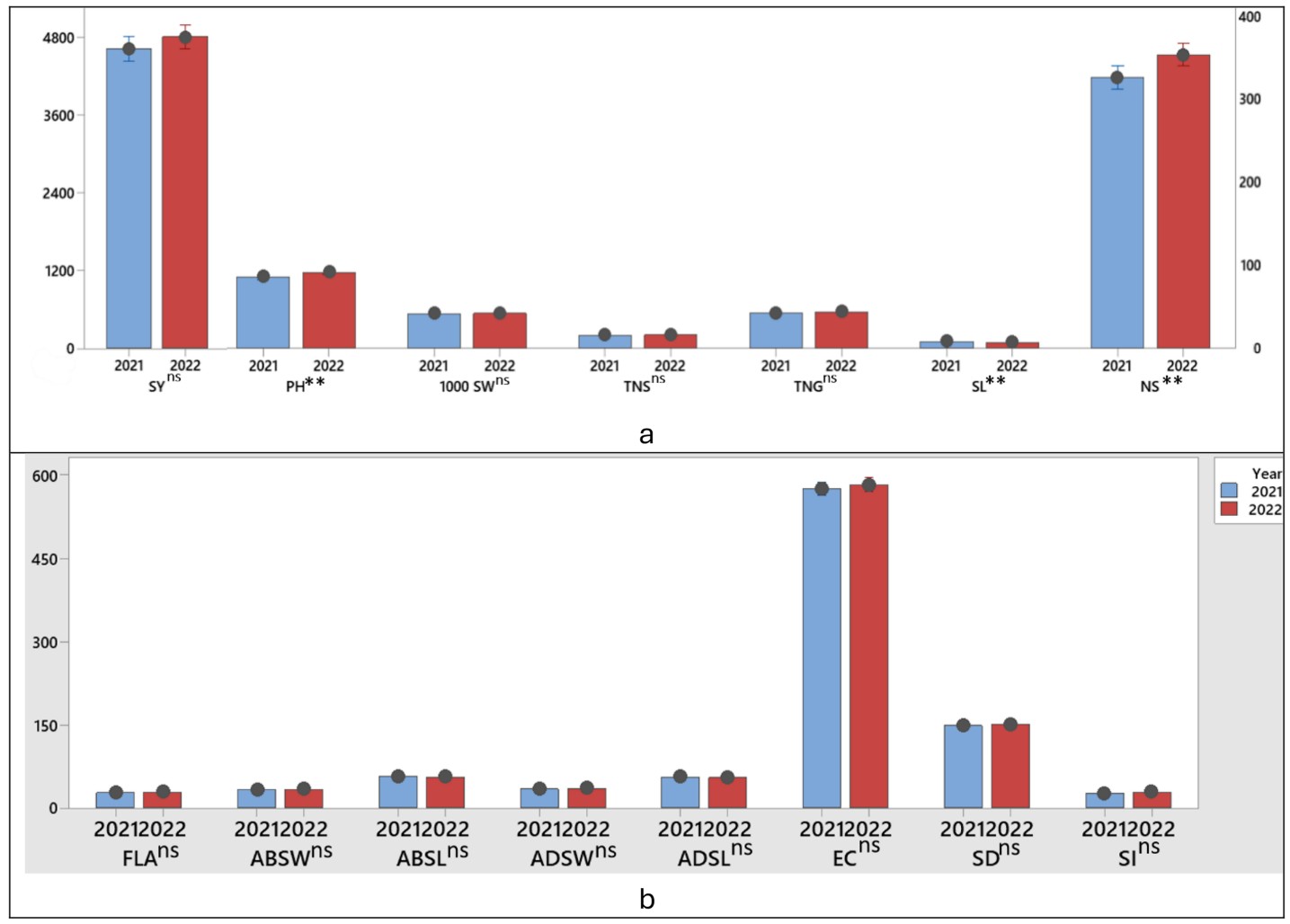

**Figure 2 T test significance control of the difference between the first and second growing season (years) between the parameters examined in wheat (A, B).** \**P ≤ 0.01; ns, non significant SY; Seed yield; PH, Plant height; TGW, 1,000 seed weight; NSS, Number of spikelets; NS, Number of spike; NGS, Number of grains; CLO, Chlorophyll content; ABSW, Abaxial stomata width; ADSW, Adaxial stomata width; FLA, Flag leaf area; ABSH, Abaxial stomatal height; ADSH, Adaxial stomatal height; SD, Stomata density; SI, Stomata index and ECN, Epidermal cell number.

2021–2022 growing seasons, respectively), the highest mean plant height was obtained from Sönmez-2001 wheat variety (94.64 cm) at first year and from Kate-A1 wheat variety (97.61 cm) at second year (Tables 2 and 3). When it comes to variety x nitrogen interaction obtained from the highest grain yield (V3), 1,000 grain weight (V8), the number of spikelet (V7) and the number of spike (V6) from N2 treatments (100 kg ha$^{-1}$) were obtained at first year (Figs. 3A–3D, respectively). N dose, varieties and variety x N dose interactions had significant effects on the 1,000 seed weight. The 1,000 seed weight increased from 5.36% as the N dose rate increased from 0 to 50 kg ha$^{-1}$ at first year (Table 3). A similar increase rate (5.79%) was also observed at second year (Table 3). According to variance analysis the effect of N doses, varieties and interaction of N with varieties on the number spikelets were significant (Tables 2 and 3, both years). When the N dose increased from 0 to 100 kg ha$^{-1}$,

**Table 2 Effects of treatment on yield and yield components of wheat in 2020–2021 (1st year) growing season ($N_n$ mean ± sd).**

| Treatment | Seed yield (kg ha$^{-1}$) | Plant height (cm) | 1,000 seed weight (g) | Number of spikelets (number/spike) | Number of grains (number/spike) | Spike length (cm) | Number of spikes (number/m$^2$) |
|---|---|---|---|---|---|---|---|
| N0 | 3,659 ± 327[c] | 72.25 ± 7.93[c] | 39.18 ± 2.67[c] | 12.63 ± 1.60[c] | 32.78 ± 4.12[c] | 7.29 ± 0.92[c] | 268.11 ± 38.58[c] |
| N1 | 4,689 ± 392[b] | 85.24 ± 10.34[b] | 41.27 ± 3.00[b] | 15.63 ± 2.34[b] | 41.85 ± 1.75[b] | 8.44 ± 1.00[b] | 313.15 ± 47.88[b] |
| N2 | 5,511 ± 409[a] | 96.94 ± 11.03[a] | 42.81 ± 3.21[a] | 19.48 ± 2.34[a] | 51.07 ± 5.10[a] | 9.35 ± 1.19[a] | 380.63 ± 35.83[a] |
| Mean | 4,619 ± 848 | 84.81 ± 14.05 | 41.08 ± 3.29 | 15.91 ± 3.51 | 41.90 ± 8.45 | 8.36 ± 3.13 | 320.63 ± 61.73 |
| V1 | 4,289 ± 775[c] | 81.25 ± 7.31[bc] | 41.09 ± 2.10[cd] | 17.11 ± 2.57[a] | 42.78 ± 8.56 | 8.22 ± 0.92[b] | 316.00 ± 66.62[ab] |
| V2 | 4,644 ± 1,041[abc] | 78.53 ± 12.76[c] | 37.36 ± 0.91[f] | 12.56 ± 1.67[c] | 42.22 ± 8.97 | 7.68 ± 1.06[b] | 313.67 ± 53.68[ab] |
| V3 | 4,900 ± 1,005[ab] | 78.63 ± 11.05[c] | 36.83 ± 1.34[f] | 15.67 ± 2.50[ab] | 42.33 ± 10.15 | 8.13 ± 1.33[b] | 310.56 ± 40.68[ab] |
| V4 | 4,444 ± 979[bc] | 86.20 ± 10.40[abc] | 42.63 ± 2.89[b] | 16.56 ± 3.64[a] | 39.67 ± 8.63 | 10.20 ± 1.14[a] | 322.00 ± 70.25[ab] |
| V5 | 4,478 ± 689[bc] | 93.26 ± 14.28[ab] | 39.43 ± 1.11[e] | 14.00 ± 3.87[bc] | 42.67 ± 7.26 | 8.30 ± 1.05[b] | 353.56 ± 30.45[ab] |
| V6 | 5,044 ± 914[a] | 76.70 ± 9.73[c] | 42.31 ± 1.01[bc] | 16.22 ± 4.18[a] | 42.22 ± 8.67 | 8.64 ± 1.67[b] | 357.44 ± 90.63[a] |
| V7 | 4,522 ± 670[abc] | 84.90 ± 11.69[abc] | 40.51 ± 2.32[de] | 17.33 ± 3.54[a] | 41.89 ± 9.64 | 8.10 ± 1.10[b] | 322.89 ± 54.99[a] |
| V8 | 4,644 ± 639[abc] | 89.20 ± 19.56[abc] | 45.92 ± 2.37[a] | 17.78 ± 2.91[a] | 41.89 ± 8.72 | 8.40 ± 1.25[b] | 280.44 ± 29.66[b] |
| V9 | 4,611 ± 948[abc] | 94.64 ± 17.99[a] | 43.70 ± 1.41[b] | 16.00 ± 4.00[a] | 41.44 ± 8.86 | 7.56 ± 0.89[b] | 309.11 ± 79.01[ab] |
| Mean | 4,619 ± 848 | 84.81 ± 14.05 | 41.08 ± 3.29 | 15.91 ± 3.51 | 41.90 ± 8.45 | 8.36 ± 3.13 | 320.63 ± 61.73 |
| N doses (n = 3) | ** | ** | ** | ** | ** | ** | ** |
| Variety (n = 9) | ** | ** | ** | ** | ns | ** | ** |
| N doses x Variety (n = 27) | ** | ns | ** | ** | ns | ns | * |

Note:
a, b; different letters in the same column indicate the difference between averages. V1, Alpu-2001; V2, Soyer-02; V3, Kate-A1; V4, Bezostaja-1; V5, Altay-2000; V6, Müfitbey; V7, Nacibey; V8, Harmankaya-99; V9, Sönmez-2001; Nitrogen doses, N0, Control; N1, 50 kg ha$^{-1}$; N2, 100 kg ha$^{-1}$; *$P \leq 0.05$; **$P \leq 0.01$; ns; non significant; sd, standard deviation.

the change in number of spikelets increased from 12.63 to 19.48 number spike$^{-1}$ in first year (Table 2) and increased from 12.74 to 19.59 number spike$^{-1}$ in the second year. When the varieties were compared individually in terms of the number spikelet, the lowest was obtained from Soyer-02 variety (12.56 number spike$^{-1}$ for 2020–2021 and 13.56 number spike$^{-1}$ for 2021–2022 growing seasons, respectively (Tables 2 and 3). While the number of grains per spike was significantly affected only by N dose at first year (Table 2), it was affected by variety and N dose at second year (Table 3). As the N dose rate increased from 0 to 50 kg ha$^{-1}$, the number of grains per spike increased 27.66% and its increase from 50 to 100 kg ha$^{-1}$ caused 22.03% growth in the number of grains per spike at first year (Table 2). Similar data was found in the second year was found similarly data (26.36% for 2020–2021and 20.93% for 2021–2022 growing seasons, respectively) (Table 3). The spike length was significantly affected by N dose, varieties and their interaction. Increasing the N dose rate from 0 to 50 kg ha$^{-1}$ resulted in increased spike length (from 7.29 to 8.44 cm for 2020–2021 and 6.29 to 7.44 cm for 2021–2022 growing seasons, respectively). As the N doses increase from 50 to 100 kg ha$^{-1}$ the rate of increase in spike length (from 8.44 to 9.35 cm for 2020–2021; 7.44 to 8.35 cm for 2021–2022 growing seasons, respectively) (Fig. 4). Among the varieties in terms of spike length, Bezostaja-1 variety stands out both years (Tables 2 and 3). N doses, varieties and their interactions had significant effects on

**Table 3 Effects of treatment on yield and yield components of wheat in 2021–2022 (2nd year) growing season (mean ± sd).**

| Treatment | Seed yield (kg ha$^{-1}$) | Plant height (cm) | 1,000 seed weight (g) | Number of spikelets (number/spike) | Number of grains (number/spike) | Spike length (cm) | Number of spikes (number/m$^2$) |
|---|---|---|---|---|---|---|---|
| N0 | 3,859 ± 295[c] | 85.66 ± 8.35[b] | 39.71 ± 2.51[c] | 12.74 ± 1.70[c] | 34.44 ± 4.68[c] | 6.29 ± 0.92[c] | 295.11 ± 38.58[c] |
| N1 | 4,867 ± 394[b] | 86.13 ± 9.93[b] | 42.01 ± 2.88[b] | 15.74 ± 2.18[b] | 43.52 ± 2.86[b] | 7.44 ± 1.00[b] | 340.15 ± 47.88[b] |
| N2 | 5,696 ± 416[a] | 99.27 ± 7.63[a] | 43.33 ± 2.91[a] | 19.59 ± 2.39[a] | 52.63 ± 5.85[a] | 8.35 ± 1.19[a] | 407.63 ± 35.83[a] |
| Mean | 4,807 ± 840 | 90.35 ± 10.67 | 41.68 ± 3.12 | 16.02 ± 3.51 | 43.53 ± 8.75 | 7.36 ± 1.33 | 347.63 ± 61.73 |
| V1 | 4,533 ± 675[c] | 93.75 ± 7.34[ab] | 40.98 ± 2.06[c] | 18.11 ± 2.57[a] | 45.78 ± 8.56[a] | 7.22 ± 0.92[b] | 343.00 ± 66.62[ab] |
| V2 | 4,833 ± 1,050[abc] | 93.23 ± 8.44[ab] | 37.85 ± 1.06[d] | 13.56 ± 1.67[d] | 45.22 ± 8.97[a] | 6.68 ± 1.06[b] | 340.67 ± 53.68[ab] |
| V3 | 5,067 ± 1,021[ab] | 97.65 ± 2.72[a] | 38.00 ± 1.36[d] | 14.67 ± 2.50[cd] | 40.33 ± 10.15[ab] | 7.13 ± 1.33[b] | 337.56 ± 40.68[ab] |
| V4 | 4,644 ± 979[bc] | 87.49 ± 9.69[ab] | 43.63 ± 2.89[ab] | 15.56 ± 3.64[bcd] | 37.67 ± 8.63[b] | 9.20 ± 1.14[a] | 349.00 ± 70.25[ab] |
| V5 | 4,667 ± 712[bc] | 93.70 ± 7.90[ab] | 40.32 ± 0.96[c] | 15.00 ± 3.87[cd] | 44.67 ± 7.26[a] | 7.30 ± 1.05[b] | 380.56 ± 30.45[a] |
| V6 | 5,222 ± 922[a] | 81.34 ± 11.21[b] | 43.27 ± 0.85[b] | 17.22 ± 4.18[ab] | 44.22 ± 8.67[ab] | 7.64 ± 1.67[b] | 384.44 ± 90.63[a] |
| V7 | 4,711 ± 653[bc] | 87.52 ± 12.72[ab] | 41.17 ± 2.54[c] | 18.33 ± 3.54[a] | 44.89 ± 9.64[a] | 7.10 ± 1.10[b] | 349.89 ± 54.99[ab] |
| V8 | 4,800 ± 634[abc] | 90.28 ± 11.34[ab] | 44.92 ± 2.28[a] | 16.78 ± 2.91[abc] | 44.89 ± 8.72[a] | 7.40 ± 1.25[b] | 307.44 ± 29.66[b] |
| V9 | 4,789 ± 949[abc] | 88.22 ± 14.98[ab] | 45.01 ± 1.62[a] | 15.00 ± 4.00[cd] | 44.11 ± 8.55[ab] | 6.56 ± 0.89[b] | 336.11 ± 79.01[ab] |
| Mean | 4,807 ± 840 | 90.35 ± 10.67 | 41.68 ± 3.12 | 16.02 ± 3.51 | 43.53 ± 8.75 | 7.36 ± 1.33 | 347.63 ± 61.73 |
| N doses ($n = 3$) | ** | ** | ** | ** | ** | ** | ** |
| Variety ($n = 9$) | ** | ** | ** | ** | ** | ** | ** |
| N doses x Variety ($n = 27$) | ** | ns | * | ** | ns | ns | * |

Note:
a, b; different letters in the same column indicate the difference between averages. V1, Alpu-2001; V2, Soyer-02; V3, Kate-A1; V4, Bezostaja-1; V5, Altay-2000; V6, Müfitbey; V7, Nacibey; V8, Harmankaya-99; V9, Sönmez-2001; Nitrogen doses, N0, Control; N1, 50 kg ha$^{-1}$; N2, 100 kg ha$^{-1}$; *$P \leq 0.05$; **$P \leq 0.01$; ns; non significant; sd, standard deviation.

the number of spikes (Tables 2, 3). The number of spikes increased from 268.11 to 380.63 (increased 41.96%) as a result of the N dose rate change from 0 to 100 kg ha$^{-1}$ at first year, while this increase rate was 38.12% at second year (Tables 2 and 3). When the varieties were compared in terms of number of spikes, the lowest was obtained from Harmankaya-99 variety and the highest was obtained from Müfitbey variety in both years (Tables 2 and 3).

## Chlorophyll content (SPAD value)

Chlorophyll content was affected by the N dose, variety, growth stage for both years. Chlorophyll content increased 19.6% for 2020–2021 and 20.55% for 2021–2022 growing season as the N dose rate increased from 0 to 50 kg ha$^{-1}$, respectively years. As the N dose rate increased from 50 to 100 kg ha$^{-1}$ the chlorophyll content increased in 15.13% and 14.64%, in the first and second years respectively. Chlorophyll content decreased as the growth period progressed in both of years. The highest chlorophyll content was obtained from variety of V7 (Nacibey) in both years (Table 4).

## Flag leaf area

N doses, varieties and their interactions had significant effects on the flag leaf area. Flag leaf area increased with increasing nitrogen doses and the highest was found at N2 treatment.

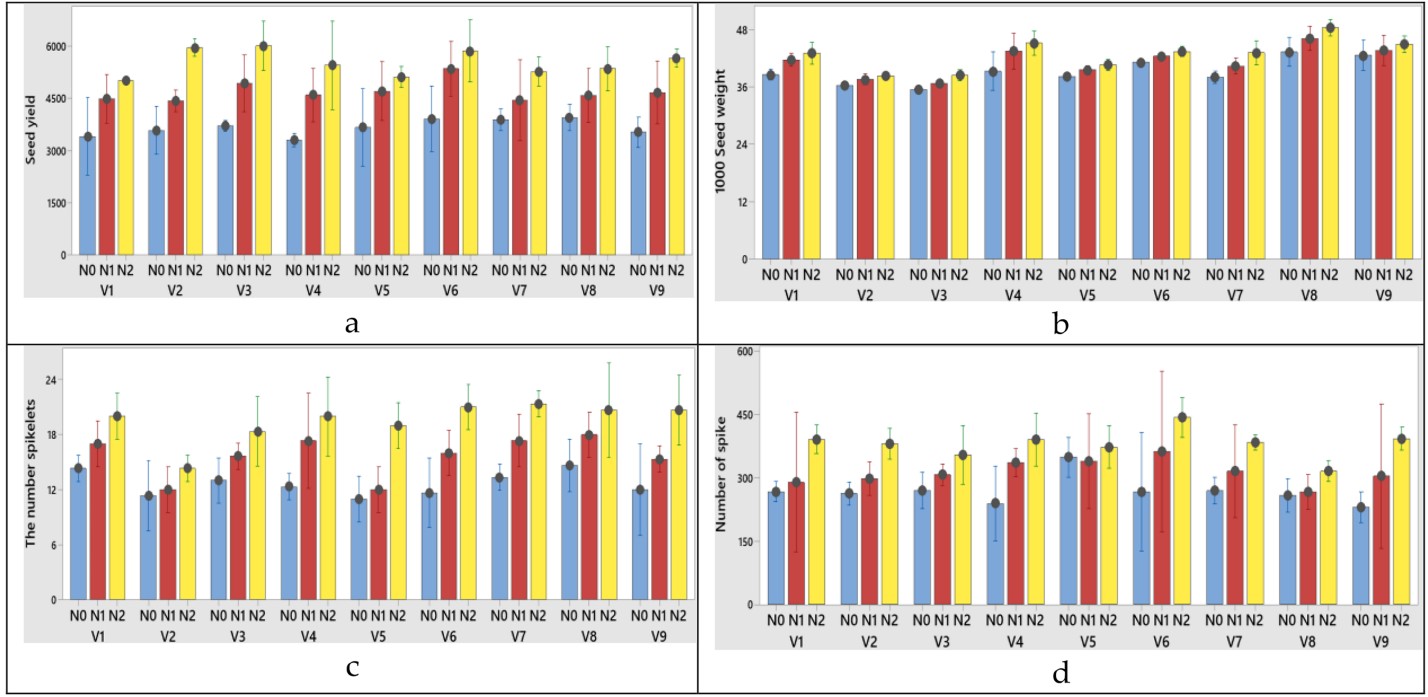

**Figure 3 (A–D) Graphs of wheat yield and yield components of the interaction of variety and nitrogen doses in the 2020–2021 (1st year) growing season.** V1, Alpu-2001; V2, Soyer-02; V3, Kate-A1; V4, Bezostaja-1; V5, Altay-2000; V6, Müfitbey; V7, Nacibey; V8, Harmankaya-99; V9, Sönmez-2001; Nitrogen doses, N0, Control; N1, 50 kg ha$^{-1}$; N2, 100 kg ha$^{-1}$.

The highest flag leaf area was determined from variety Bezostaja-1, while the lowest flag leaf area index was obtained from variety Soyer-02 in both years (Tables 5 and 6).

As the N dose rate increased from 0 to 50 kg ha$^{-1}$, the flag leaf area increased from 25.31 to 27.90 cm$^2$ (10.23%), and similarly as the N dose rate increased from 50 to 100 kg ha$^{-1}$, the flag leaf area increased 10.21% at first year. In the second year, this ratio was determined between 11.04% and 17.18%, in relation to N doses respectively. The variation of flag leaf area index of the varieties was found between 21.74–36.22 cm$^2$ (first year) and between 21.54–35.54 cm$^2$ (second year) (Tables 5 and 6).

The highest flag leaf area (V4), N2 treatment (100 kg ha$^{-1}$) was obtained from variety x nitrogen interaction at both years (Figs. 5A, 6A).

## Characteristics of stomata

The changes of N dose had significant effects on abaxial and adaxial stomata width and length, epidermal cell, stomata density and stomata index in this study. There was an increase in these parameters with the increase in nitrogen doses.

Among the varieties abaxial width changed between 30.06 and 35.95 μm in the first year; 31.06 and 36.95 μm in the second year. Abaxial stomata length was obtained from 54.02 and 61.45 μm for 2020–2021; 53.02 and 60.45 μm for 2021–2022 growing season, respectively years (Tables 5 and 6). When the 1st and 2nd year abaxial stomata width (μm), abaxial stomata length (μm), adaxial stomata width (μm), adaxial stomata length (μm), epidermal cell (no/mm$^2$), stomata density (no/mm$^2$) and stomata index (%) data were

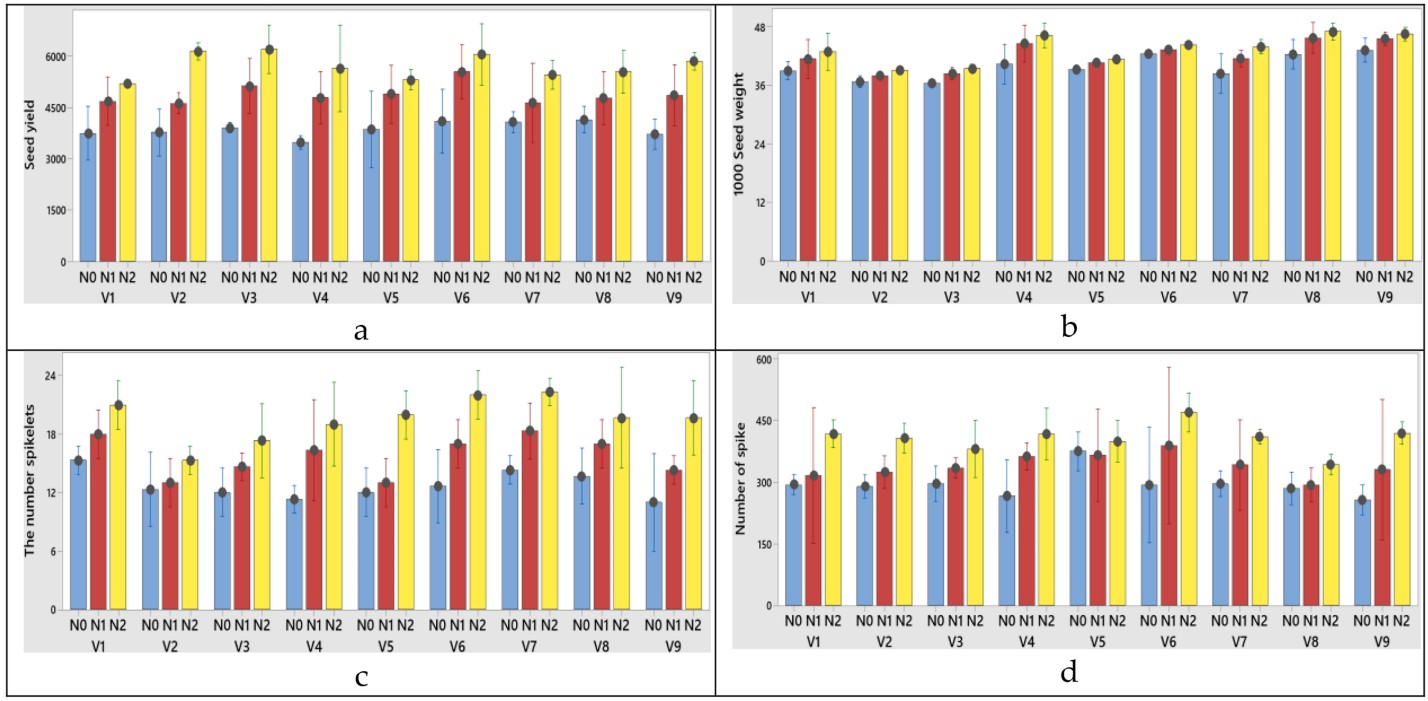

**Figure 4 (A-D) Graphs of wheat yield and yield components of the interaction of variety and nitrogen doses in the 2021–2022 (2nd year) growing season.** V1, Alpu-2001; V2, Soyer-02; V3, Kate-A1; V4, Bezostaja-1; V5, Altay- 2000; V6, Müfitbey; V7, Nacibey; V8, Harmankaya-99; V9, Sönmez-2001; Nitrogen doses, N0, Control; N1, 50 kg ha$^{-1}$; N2, 100 kg ha$^{-1}$.

examined, the variety x nitrogen dose interaction was found to be significant in all parameters except abaxial stomata length (μm) and adaxial stomata length (μm). When the 1st year data is examined the highest abaxial stomata width was obtained from variety Nacibey and 100 kg ha$^{-1}$ N dose interactions (39.27 μm) and the lowest adaxial stomata width was obtained from Bezostaja-1variety and control (0 kg ha$^{-1}$ N) dose interactions (22.77 μm). The highest abaxial stomata length was obtained from variety Harmankaya-99 and 100 kg ha$^{-1}$ N dose (62.46 μm) and the lowest abaxial stomata width was obtained from Nacibey variety and control (0 kg ha$^{-1}$ N) dose interactions (49.53 μm). N doses had significant effects on the abaxial and adaxial stomata length in the wheat. As N doses increase, abaxial and adaxial stomata length also increases respectively (Fig. 5). N doses, varieties and N dose x variety interactions had significant effects on the adaxial stomata width. The highest adaxial stomata width was obtained from Nacibey variety and 100 kg ha$^{-1}$ N dose interactions (41.77 μm) and the lowest adaxial stomata width was obtained from variety Bezostaja 1 and control (0 kg ha$^{-1}$ N) dose interactions (25.27 μm). N doses had significant effects on the adaxial stomata width in the wheat. As N doses increase stomata width also increases respectively (Fig. 5). The highest abaxial stomata width (second year) was obtained from Nacibey variety and 100 kg ha$^{-1}$ N dose interactions (40.27 μm) and the lowest adaxial stomata width was obtained from variety Bezostaja 1 and control (0 kg ha$^{-1}$ N) dose interactions (23.77 μm). The highest abaxial stomata length was obtained from Harmankaya-99 variety and 100 kg ha$^{-1}$ N dose (61.46 μm) and the lowest abaxial stomata width was obtained from Nacibey variety and control

**Table 4 Effects of Treatment on chlorophyll content of wheat in 2021–2022 and 2021–2022 growing season (mean ± sd).**

| 2020–2021 growing season | | 2021–2022 growing season | |
| --- | --- | --- | --- |
| Treatment | Chlorophyll (SPAD value) | Treatment | Chlorophyll (SPAD value) |
| N0 | 22.96 ± 5.66[c] | N0 | 23.40 ± 5.69[c] |
| N1 | 27.48 ± 5.44[b] | N1 | 28.21 ± 5.54[b] |
| N2 | 31.64 ± 6.27[a] | N2 | 32.34 ± 6.39[a] |
| Mean | 27.35 ± 6.75 | Mean | 27.98 ± 6.88 |
| GS1 | 29.86 ± 6.80[a] | GS1 | 30.58 ± 6.99[a] |
| GS2 | 27.52 ± 6.34[a] | GS2 | 28.00 ± 6.41[a] |
| GS3 | 24.69 ± 6.25[b] | GS3 | 25.38 ± 6.39[b] |
| Mean | 27.35 ± 6.75 | Mean | 27.98 ± 6.88 |
| N doses (*n* = 3) | ** | N doses (*n* = 3) | ** |
| Growth Stage (*n* = 3) | ** | Growth stage (*n* = 3) | ** |
| N doses x Growth Stage (*n* = 9) | ns | N doses x Growth Stage (*n* = 9) | ns |
| N0 | 22.96 ± 4.82[c] | N0 | 24.92 ± 4.97[c] |
| N1 | 27.48 ± 4.72[b] | N1 | 29.20 ± 4.93[b] |
| N2 | 31.64 ± 5.40[a] | N2 | 33.58 ± 5.12[a] |
| Mean | 27.36 ± 4.98 | Mean | 29.23 ± 5.01 |
| V1 | 21.41 ± 4.25[c] | V1 | 22.97 ± 4.92[c] |
| V2 | 20.75 ± 5.48[c] | V2 | 22.82 ± 5.64[c] |
| V3 | 30.84 ± 5.58[a] | V3 | 32.64 ± 4.99[a] |
| V4 | 29.34 ± 5.00[ab] | V4 | 31.09 ± 4.75[ab] |
| V5 | 28.01 ± 3.42[ab] | V5 | 30.14 ± 2.84[ab] |
| V6 | 24.03 ± 2.31[bc] | V6 | 26.13 ± 2.48[bc] |
| V7 | 33.20 ± 5.59[a] | V7 | 34.98 ± 5.60[a] |
| V8 | 29.04 ± 3.47[ab] | V8 | 30.79 ± 3.69[ab] |
| V9 | 29.59 ± 6.19[a] | V9 | 31.56 ± 6.56[a] |
| Mean | 27.36 ± 4.98 | Mean | 29.23 ± 5.01t |
| N doses (*n* = 3) | ** | N doses (*n* = 3) | ** |
| Variety (*n* = 9) | ** | Variety (*n* = 9) | ** |
| N doses x Variety (*n* = 27) | ns | N doses x Variety (*n* = 27) | ns |

Note:
a, b; different letters in the same column indicate the difference between averages. V1, Alpu-2001; V2, Soyer-02; V3, Kate-A1; V4, Bezostaja-1; V5, Altay-2000; V6, Müfitbey; V7, Nacibey; V8, Harmankaya-99; V9, Sönmez-2001; Nitrogen doses, N0, Control; N1, 50 kg ha$^{-1}$; N2, 100 kg ha$^{-1}$; **$P \leq 0.01$; ns, non significant; sd, standard deviation; GS, Growth stage.

(0 kg ha$^{-1}$ N) dose interactions (48.53 μm). N doses had significant effects on the abaxial and adaxial stomata length in the wheat. As N doses increase, abaxial and adaxial stomata length also increases respectively (Fig. 6). N doses, varieties and N dose x variety interactions had significant effects on the adaxial stomata width. The highest adaxial stomata width was obtained from Nacibey variety and 100 kg ha$^{-1}$ N dose interactions (41.77 μm) and the lowest adaxial stomata width was obtained from Bezostaja-1 variety and control (0 kg ha$^{-1}$ N) dose interactions (25.27 μm). N doses had significant effects on the adaxial stomata width in the wheat. As N doses increase stomata width also increases respectively (Fig. 6). In the 1$^{st}$ and 2$^{nd}$ years, N dose x variety interactions had no

**Table 5 Effects of treatment on flag leaf area and stomatal characteristics of wheat in 2020–2021 (1st year) growing season ($N_n$ mean ± sd).**

| Nitrogen dose | Flag leaf area (cm²) | Abaxial stomata width (µm) | Abaxial stomata length (µm) | Adaxial stomata width (µm) | Adaxial stomata length (µm) | Epidermal cell (no./mm²) | Stomata density (no./mm²) | Stomata index (%) |
|---|---|---|---|---|---|---|---|---|
| N0 | 25.31 ± 4.99[c] | 28.94 ± 4.15[c] | 53.49 ± 3.73[c] | 30.44 ± 4.15[c] | 52.36 ± 4.19[c] | 539.59 ± 55.04[c] | 138.44 ± 11.50[c] | 20.78 ± 4.41[c] |
| N1 | 27.90 ± 5.50[b] | 32.89 ± 3.02[b] | 57.40 ± 3.14[b] | 34.39 ± 3.02[b] | 55.90 ± 3.14[b] | 584.11 ± 48.10[b] | 149.22 ± 12.35[b] | 26.11 ± 6.41[b] |
| N2 | 30.75 ± 7.02[a] | 36.79 ± 1.89[a] | 60.09 ± 2.87[a] | 38.29 ± 1.89[a] | 58.59 ± 2.87[a] | 604.41 ± 47.01[a] | 158.15 ± 11.47[a] | 32.11 ± 6.82[a] |
| Mean | 27.98 ± 6.23 | 32.87 ± 4.48 | 56.99 ± 4.22 | 34.37 ± 4.48 | 55.61 ± 4.26 | 576.03 ± 56.53 | 148.60 ± 14.17 | 26.33 ± 7.51 |
| V1 | 21.74 ± 1.15[d] | 32.28 ± 3.43[abc] | 59.72 ± 1.80[ab] | 33.78 ± 3.43[abc] | 58.22 ± 1.80[ab] | 622.00 ± 14.71[b] | 154.78 ± 7.90[bc] | 31.89 ± 7.83[b] |
| V2 | 23.30 ± 1.50[d] | 33.63 ± 3.43[abc] | 57.87 ± 3.41[abc] | 35.13 ± 3.43[abc] | 56.37 ± 3.41[abc] | 591.78 ± 53.11[c] | 151.00 ± 12.85[d] | 30.56 ± 9.48[b] |
| V3 | 23.30 ± 1.21[d] | 34.94 ± 2.44[ab] | 54.56 ± 3.53[c] | 36.44 ± 2.44[ab] | 53.06 ± 3.53[c] | 574.56 ± 40.98[d] | 142.67 ± 9.96[e] | 24.89 ± 6.97[d] |
| V4 | 36.22 ± 4.85[a] | 30.06 ± 6.07[c] | 56.49 ± 4.15[bc] | 31.56 ± 6.07[c] | 54.99 ± 4.15[bb] | 513.67 ± 15.58[f] | 133.00 ± 9.45[g] | 19.22 ± 3.31[f] |
| V5 | 32.08 ± 1.54[b] | 31.73 ± 5.40[bc] | 58.14 ± 4.34[abc] | 33.23 ± 5.40[bc] | 57.76 ± 4.23[ab] | 611.56 ± 15.81[b] | 155.33 ± 5.20[b] | 27.89 ± 2.98[c] |
| V6 | 26.17 ± 1.03[c] | 30.13 ± 7.12[c] | 54.02 ± 3.68[c] | 31.63 ± 7.12[c] | 52.52 ± 3.68[c] | 500.33 ± 48.96[f] | 137.44 ± 6.69[f] | 21.11 ± 3.82[e] |
| V7 | 33.83 ± 1.71[ab] | 35.95 ± 3.02[a] | 55.37 ± 5.13[c] | 37.45 ± 3.02[a] | 53.87 ± 5.13[bc] | 583.78 ± 30.12[cd] | 152.56 ± 10.01[cd] | 27.33 ± 4.80[c] |
| V8 | 32.76 ± 6.84[b] | 32.96 ± 1.73[abc] | 61.45 ± 2.35[a] | 34.46 ± 1.73[abc] | 59.95 ± 2.35[a] | 647.00 ± 30.76[a] | 171.56 ± 9.53[a] | 34.78 ± 3.87[a] |
| V9 | 22.50 ± 4.16[d] | 34.19 ± 2.40[ab] | 55.33 ± 3.71[c] | 35.69 ± 2.40[ab] | 53.83 ± 3.71[bc] | 539.67 ± 19.69[e] | 139.11 ± 8.15[f] | 19.33 ± 2.96[f] |
| Mean | 27.98 ± 6.23 | 32.87 ± 4.48 | 56.99 ± 4.22 | 34.37 ± 4.48 | 55.61 ± 4.26 | 576.03 ± 56.53 | 148.60 ± 14.17 | 26.33 ± 7.51 |
| N doses (n = 27) | ** | ** | ** | ** | ** | ** | ** | ** |
| Variety (n = 9) | ** | ** | ** | ** | ** | ** | ** | ** |
| N doses x Variety (n = 3) | ** | ** | ns | ** | ns | ** | ** | ** |

**Note:**
a, b; different letters in the same column indicate the difference between averages. V1, Alpu-2001; V2, Soyer-02; V3, Kate-A1; V4, Bezostaja-1; V5, Altay-2000; V6, Müfitbey; V7, Nacibey; V8, Harmankaya-99; V9, Sönmez-2001; Nitrogen doses, N0, Control; N1, 50 kg h⁻¹; N2, 100 kg ha⁻¹; **$P \leq 0.01$; ns, non significant; sd, standard deviation.

significant effect on abaxial and adaxial stomata length. However, abaxial and adaxial stomata lengths changed depending on N doses and varieties. In the 1st year data, as N doses increased, abaxial and adaxial stomata length values increased by 12.3% from 53.49 to 60.09 and 11.8% from 52.36 to 58.59, respectively (Table 5). As N doses increased, abaxial and adaxial stomata length values increased 11.7% from 52.86 to 59.09 and 12.1% from 51.36 to 57.59, respectively (2nd year). In the 1st and 2nd year data, the highest abaxial and adaxial stomata lengths were obtained from the Harmankaya-99 variety (61.45, 60.45 µm and 59.95, 58.95 µm, respectively) (Tables 5 and 6). In the 1st and 2nd years, N dose x variety interactions had a significant effect on epidermal cell (no/mm²), stomatal density (no/mm²), and stomatal index (%). In the 1st year data, the epidermal cell value was highest in the Harmankaya-99 (from 614 to 679.33, 10.6%) variety, and the lowest in the Müfitbey (from 439 to 548.33, 24.9%) variety (Fig. 5). In the 2nd year data, the epidermal cell value was similar to the 1st year data, and the highest value was in the Harmankaya-99 (from 621 to 686.33, 10.5%) variety, and the lowest value was in the Müfitbey (from 446 to 555.33, 24.5%) variety (Fig. 6). In the 1st year data, the stomata density value was lowest in the Müfitbey (from 144 to 129, 10.6%) variety and highest in the Sönmez-2001 (from 148

**Table 6 Effects of treatment on flag leaf area and stomatal characteristics of wheat in 2021–2022 (2nd year) growing season ($N_n$ mean ± sd).**

| Nitrogen dose | Flag leaf area (cm$^2$) | Abaxial stomata width (µm) | Abaxial stomata length (µm) | Adaxial stomata width (µm) | Adaxial stomata length (µm) | Epidermal cell (no./mm$^2$) | Stomata density (no./mm$^2$) | Stomata index (%) |
|---|---|---|---|---|---|---|---|---|
| N0 | 25.26 ± 5.04[c] | 29.94 ± 4.15[c] | 52.86 ± 4.19[c] | 31.44 ± 4.15[c] | 51.36 ± 4.19[c] | 546.59 ± 55.04[c] | 140.44 ± 11.50[c] | 22.78 ± 4.41[c] |
| N1 | 28.05 ± 5.09[b] | 33.89 ± 3.02[b] | 56.40 ± 3.14[b] | 35.39 ± 3.02[b] | 54.90 ± 3.14[b] | 591.11 ± 48.10[b] | 151.22 ± 12.35[b] | 28.11 ± 6.41[b] |
| N2 | 32.87 ± 8.07[a] | 37.79 ± 1.89[a] | 59.09 ± 2.87[a] | 39.29 ± 1.89[a] | 57.59 ± 2.87[a] | 611.41 ± 47.01[a] | 160.15 ± 11.47[a] | 34.11 ± 6.82[a] |
| Mean | 28.72 ± 6.91 | 33.87 ± 4.48 | 56.11 ± 4.26 | 35.37 ± 4.48 | 54.61 ± 4.26 | 583.03 ± 56.53 | 150.60 ± 14.17 | 28.33 ± 7.51 |
| V1 | 21.54 ± 1.33[e] | 33.28 ± 3.43[abc] | 58.72 ± 1.80[ab] | 34.78 ± 3.43[abc] | 57.22 ± 1.80[ab] | 629.00 ± 14.71[b] | 156.78 ± 7.90[bc] | 33.89 ± 7.83[b] |
| V2 | 22.62 ± 1.19[e] | 34.63 ± 3.43[abc] | 56.87 ± 3.41[abc] | 36.13 ± 3.43[abc] | 55.37 ± 3.41[abc] | 598.78 ± 53.11[c] | 153.00 ± 12.85[d] | 32.56 ± 9.48[b] |
| V3 | 22.94 ± 1.02[de] | 35.94 ± 2.44[ab] | 53.56 ± 3.53[c] | 37.44 ± 2.44[ab] | 52.06 ± 3.53[c] | 581.56 ± 40.98[d] | 144.67 ± 9.96[e] | 26.89 ± 6.97[d] |
| V4 | 35.39 ± 5.24[a] | 31.06 ± 6.07[c] | 55.49 ± 4.15[bc] | 32.56 ± 6.07[c] | 53.99 ± 4.15[bc] | 520.67 ± 15.58[f] | 135.00 ± 9.45[g] | 21.22 ± 3.31[f] |
| V5 | 35.54 ± 2.53[a] | 32.73 ± 5.40[bc] | 58.26 ± 4.23[ab] | 34.23 ± 5.40[bc] | 56.76 ± 4.23[ab] | 618.56 ± 15.81[b] | 157.33 ± 5.20[b] | 29.89 ± 2.98[c] |
| V6 | 25.81 ± 1.28[d] | 31.13 ± 7.12[c] | 53.02 ± 3.68[c] | 32.63 ± 7.12[c] | 51.52 ± 3.68[c] | 507.33 ± 48.96[f] | 139.44 ± 6.69[f] | 23.11 ± 3.82[e] |
| V7 | 34.38 ± 5.82[ab] | 36.95 ± 3.02[a] | 54.37 ± 5.13[bc] | 38.45 ± 3.02[a] | 52.87 ± 5.13[bc] | 590.78 ± 30.12[cd] | 154.56 ± 10.01[cd] | 29.33 ± 4.80[c] |
| V8 | 31.30 ± 7.48[bc] | 33.96 ± 1.73[abc] | 60.45 ± 2.35[a] | 35.46 ± 1.73[abc] | 58.95 ± 2.35[a] | 654.00 ± 30.76[a] | 173.56 ± 9.53[a] | 36.78 ± 3.87[a] |
| V9 | 29.00 ± 7.48[c] | 35.19 ± 2.40[ab] | 54.33 ± 3.71[bc] | 36.69 ± 2.40[ab] | 52.83 ± 3.71[bc] | 546.67 ± 19.69[e] | 141.11 ± 8.15[f] | 21.33 ± 2.96[f] |
| Mean | 28.72 ± 6.91 | 33.87 ± 4.48 | 56.11 ± 4.26 | 35.37 ± 4.48 | 54.61 ± 4.26 | 583.03 ± 56.53 | 150.60 ± 14.17 | 28.33 ± 7.51 |
| N doses (n = 27) | ** | ** | ** | ** | ** | ** | ** | ** |
| Variety (n = 9) | ** | ** | ** | ** | ** | ** | ** | ** |
| N doses x Variety (n = 3) | ** | ** | ns | ** | ns | ** | ** | ** |

**Note:**
a, b; different letters in the same column indicate the difference between averages. V1, Alpu-2001; V2, Soyer-02; V3, Kate-A1; V4, Bezostaja-1; V5, Altay-2000; V6, Müfitbey; V7, Nacibey; V8, Harmankaya-99; V9, Sönmez-2001; Nitrogen doses, N0, Control; N1, 50 kg ha$^{-1}$; N2, 100 kg ha$^{-1}$; **$P \leq 0.01$; ns, non significant; sd, standard deviation.

to 129) variety (Fig. 5). In the 2nd year data, the stomata density value was similar to the 1st year data and the lowest value was highest in the Müfitbey (from 146 to 131) variety and highest in the Sönmez-2001 (from 150 to 131) variety (Fig. 6). In the 1st year data, the highest stomata index value was seen in the varieties Soyer-02, Alpu-2001, and Harmankaya-99 (from 28.66 to 40, from 23.33 to 41, and from 30 to 38.66, respectively) in the same statistical group (Fig. 5). In the 2nd year data, the stomata index value was similar to the 1st year data and the highest values were reached in the varieties Soyer-02, Alpu-2001, and Harmankaya-99 (from 20.66 to 42, from 25 to 43, and from 32 to 40.66, respectively) in the same statistical group (Fig. 5). In addition, all parameters except plant height and spike length values were not statistically affected by the years and no significant difference was observed between the years (Table 6).

## PRINCIPAL COMPONENT ANALYSIS (PCA)

A PCA was performed to explore the relationships among measured variables and assess how nitrogen doses (N0, N1, N2) influence the dataset of nine varieties of wheat. The PCA

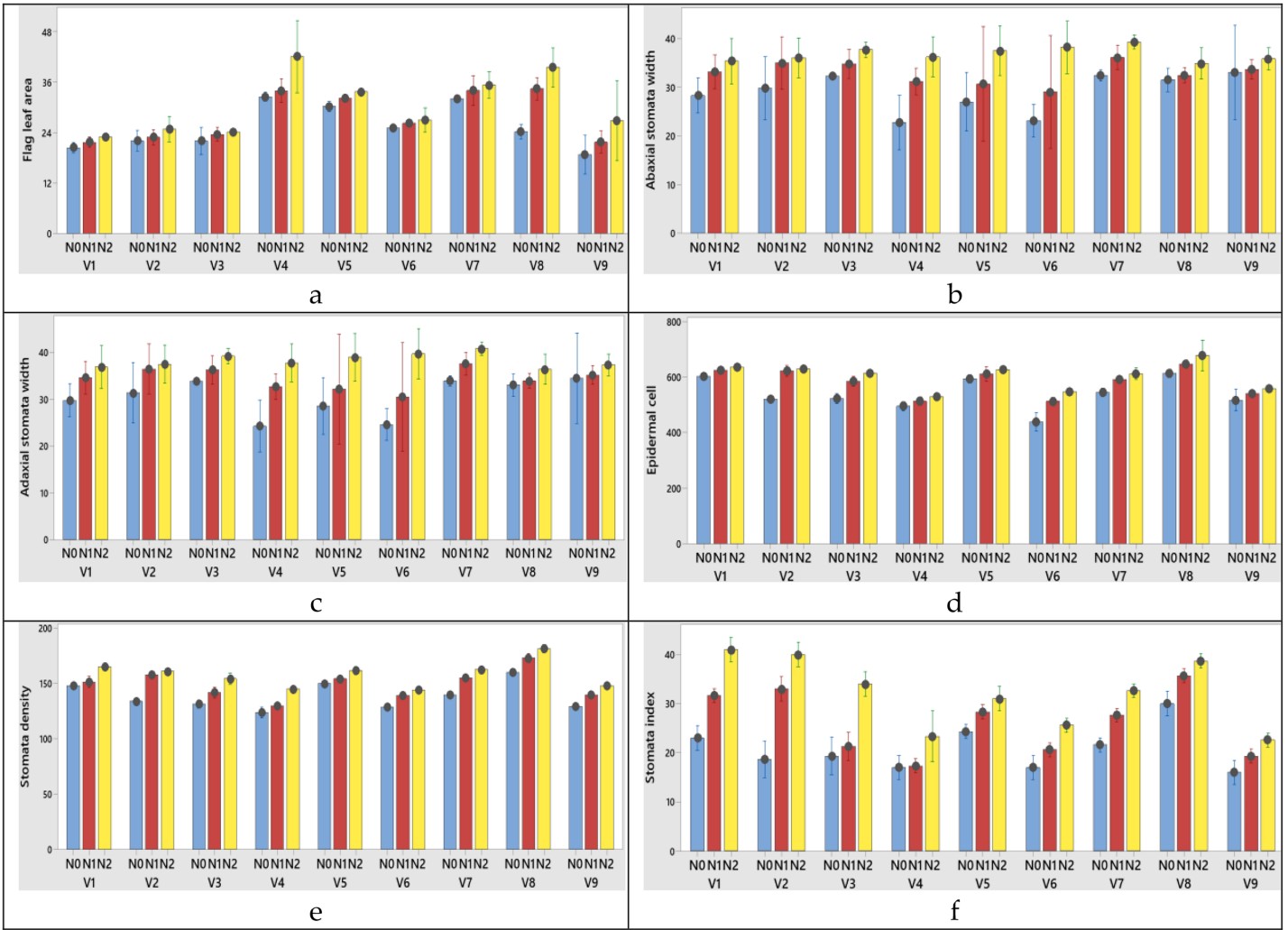

**Figure 5** (A–F) Graphs of wheat flag leaf area and stomatal characteristics of the interaction of variety and nitrogen doses in the 2020–2021 (1$^{st}$ year) growing season. V1, Alpu-2001; V2, Soyer-02; V3, Kate-A1; V4, Bezostaja-1; V5, Altay-2000; V6, Müfitbey; V7, Nacibey; V8, Harmankaya-99; V9, Sönmez-2001; Nitrogen doses, N0, Control; N1, 50 kg ha$^{-1}$; N2, 100 kg ha$^{-1}$.

biplot revealed a clear separation among the nitrogen dose groups, reflecting differences in their multidimensional profiles.

The first and two dimensions (Dim1 and Dim2) accounted for a significant proportion of the total variance in the dataset, with Dim1 explaining 50.5% and Dim2 explaining 13.3%. Together, they captured the major trends within the data. According to the principal component analysis, the traits examined in the study were grouped into two main groups in terms of colour and length. The first group (Dim 1) consisted of seed yield (SY), plant height (PH), thousand grain weight (TGW), number of spikelets (NSS), number of spike (NS), number of grains (NGS), chlorophyll content (CLO), abaxial stomata width (ABSW), adaxial stomata width (ADSW) and flag leaf area (FLA) components that demonstrated strong positive intercorrelations. These features are associated with productivity and photosynthetic capacity, both of which are directly influenced by nitrogen availability. The second group (Dim 2) consisted of abaxial

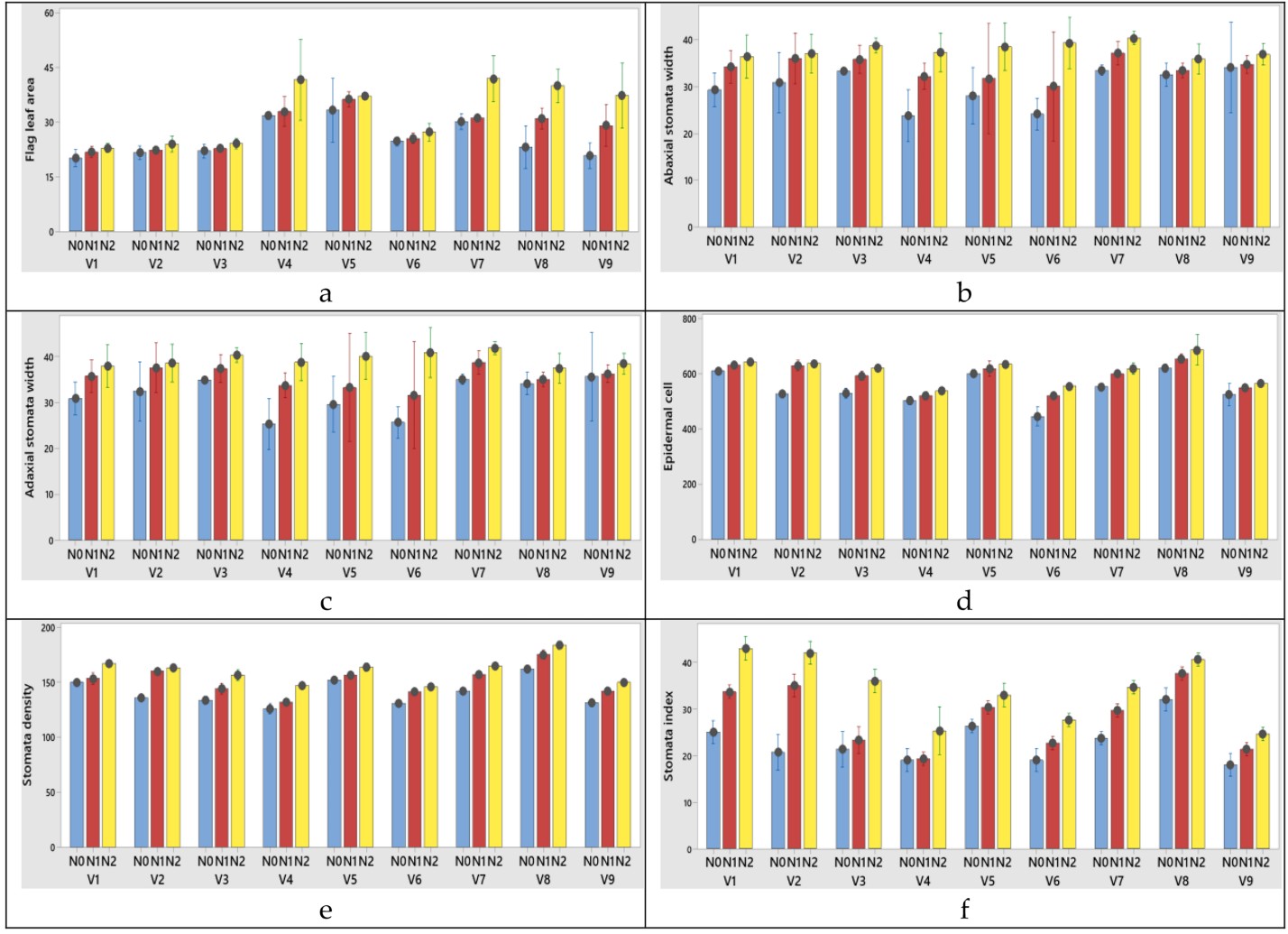

**Figure 6 (A–F) Graphs of wheat flag leaf area and stomatal characteristics of the interaction of variety and nitrogen doses in the 2021–2022 (2nd year) growing season.** V1, Alpu-2001; V2, Soyer-02; V3, Kate-A1; V4, Bezostaja-1; V5, Altay-2000; V6, Müfitbey; V7, Nacibey; V8, Harmankaya-99; V9, Sönmez-2001; Nitrogen doses, N0, Control; N1, 50 kg ha$^{-1}$; N2, 100 kg ha$^{-1}$.

stomatal height (ABSH), adaxial stomatal height (ADSH), stoma density (SD), stoma index (SI) and epidermal cell number (ECN) that features represents stomatal and cellular characteristics. These components primarily aligned along Dim2 and has shown strong positive intercorrelations (Fig. 7). All the analyzed features showed nearly positive correlations while epidermal cell number (ECN) and flag leaf area (FLA) exhibited non correlation.

The nitrogen dose groups were distinctly separated along Dim1 and Dim2. The control group (N0) clustered toward the negative end of Dim1, suggesting lower values of variables strongly associated with this dimension. This indicates a limited growth response in the absence of nitrogen. In contrast, the 100 kg ha$^{-1}$ N dose group (N2) positioned toward the positive end of Dim2, indicating higher growth and yield parameters. The 50 kg ha$^{-1}$ N dose group (N1) occupied an intermediate position, bridging the control and 100 kg ha$^{-1}$ N groups, showed moderate increases in features (Fig. 8). Interestingly, stomatal traits

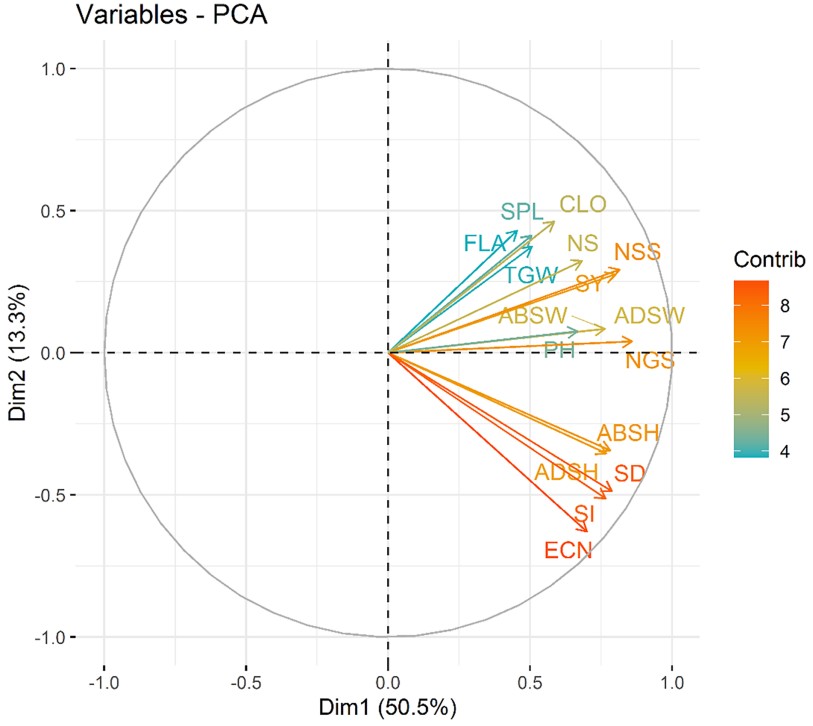

**Figure 7** **Variable contribution plot for the first two dimensions (Dim1 and Dim2) from the PCA analysis.** The length, direction and the color of the arrows in the PCA represents the contribution of each variable to the first two principal components. SY, Seed yield; PH, Plant height; TGW, 1,000 seed weight; NSS, Number of spikelets; NS, Number of spike; NGS, Number of grains; CLO, Chlorophyll content; ABSW, Abaxial stomata width; ADSW, Adaxial stomata width; FLA, Flag leaf area; ABSH, Abaxial stomatal height; ADSH, Adaxial stomatal height; SD, Stomata density; SI, Stomata index and ECN, Epidermal cell number.

(Dim2) show less dramatic shifts across N levels, suggesting that these features are less directly affected by N dosage. The results highlight the dose-dependent effects of N on plant development and productivity parameters.

The color gradient in Fig. 9 represents the contribution of each variety to the principal components. Varieties with higher contributions, such as Bezostaja-1 (V4) and Sönmez-2001 (V9), are more influential in defining the axes of variation. This highlights the importance of varietal differences in the effectiveness of nitrogen application.

## DISCUSSION

There is a relationship between the N dose and the yield (*Paul et al., 2017*; *Luo et al., 2019*). As we found in the present study, seed yield and agronomic traits had positive relations with N dose treatments. Many studies have reported that increase of N also increase yield and agronomic traits (*Fang et al., 2018*; *Lollato et al., 2019*; *Liu, Liao & Liu, 2021*). In our study, as N doses increased, all yield and morphological parameters increased as well. High N dose stimulates the development of vegetative organs and reduces the amount of assimilation carried to the spikes (*Sun et al., 2019*). While the high values obtained in the examined properties were obtained from the highest N dose, the percentage of increase occurring in the increase from control (0 kg ha$^{-1}$) to 50 kg ha$^{-1}$ was greater than the

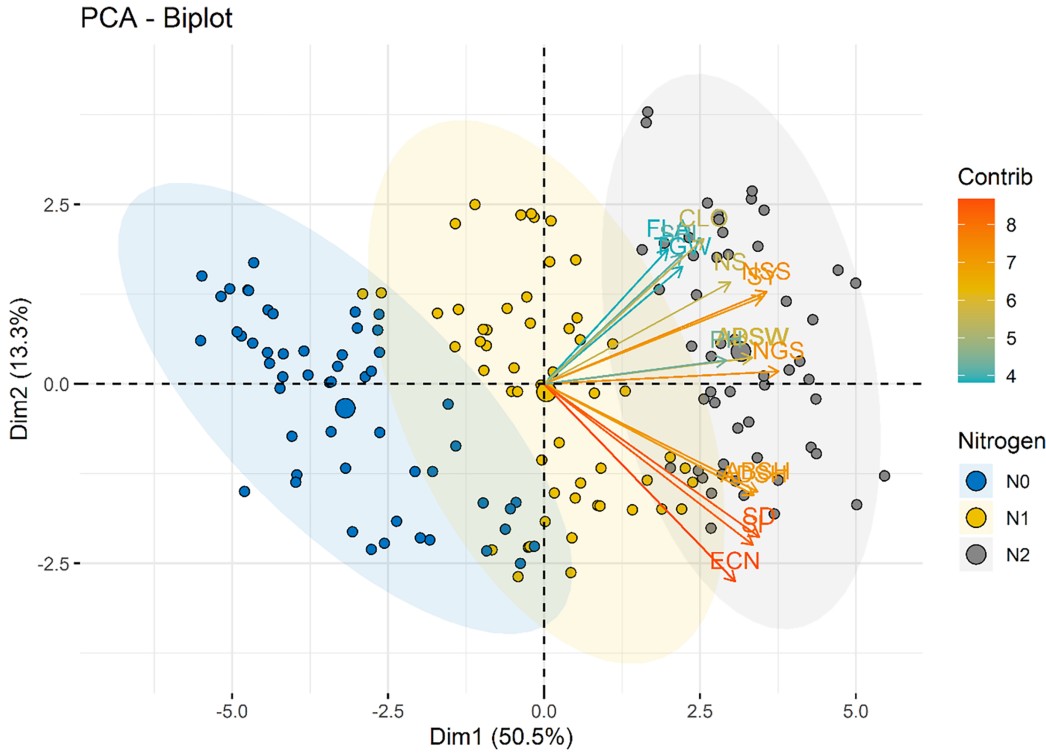

**Figure 8 PCA biplot illustrating the distribution of individuals across the first two dimensions (Dim1 and Dim2).** Nitrogen dose groups (N0, Control; N1, 50 kg ha$^{-1}$; N2, 100 kg ha$^{-1}$) are color-coded, with ellipses representing 95% confidence intervals for each group. Variable loadings are depicted as arrows, highlighting their contributions to Dim1 and Dim2 and the group separation.

percentage of increase occurring in the increase from 50 to 100 kg ha$^{-1}$. The highest seed yield was obtained from 100 kg ha$^{-1}$ nitrogen application in our study. *Kayan et al. (2020)* determined 100 kg ha$^{-1}$ as the most suitable N dose in terms of seed yield and N efficiency among N doses. The rapid response of seed yield to N increases at low N levels is due to the increase in tiller formation and the resulting increase in seed number when the soil N amount is insufficient for maximum yield (*Hoogmoed et al., 2018*). Despite the potential increase in plant N concentration, subsequent slow growth phases indicate that increases in N uptake per plant beyond the maximum yield response point result in low improvement in yield per plant (*Xu et al., 2018*). Different N dose could improve the photosynthetic capacity by regulating the plant height, thereby increasing production (*Lollato et al., 2019*). As a result of the correlation, it was determined that seed yield was highly positively correlated with plant height, number of spikelets per spikes, number of seeds per spike and 1,000 seeds weight with increase N treatment in both years in our study (Table 2). *Xue et al. (2006)* reported that N dose improved yield by increasing seed number. That the improving number of spikes and number of seeds per spike may be the primary factors for increasing yield (*Si et al., 2020*). *Saleem et al. (2010)* stated that more studies are required to identify wheat cultivars that maintain high seed yield with lower N dose requirements. That researchers found seed yield increased significantly and

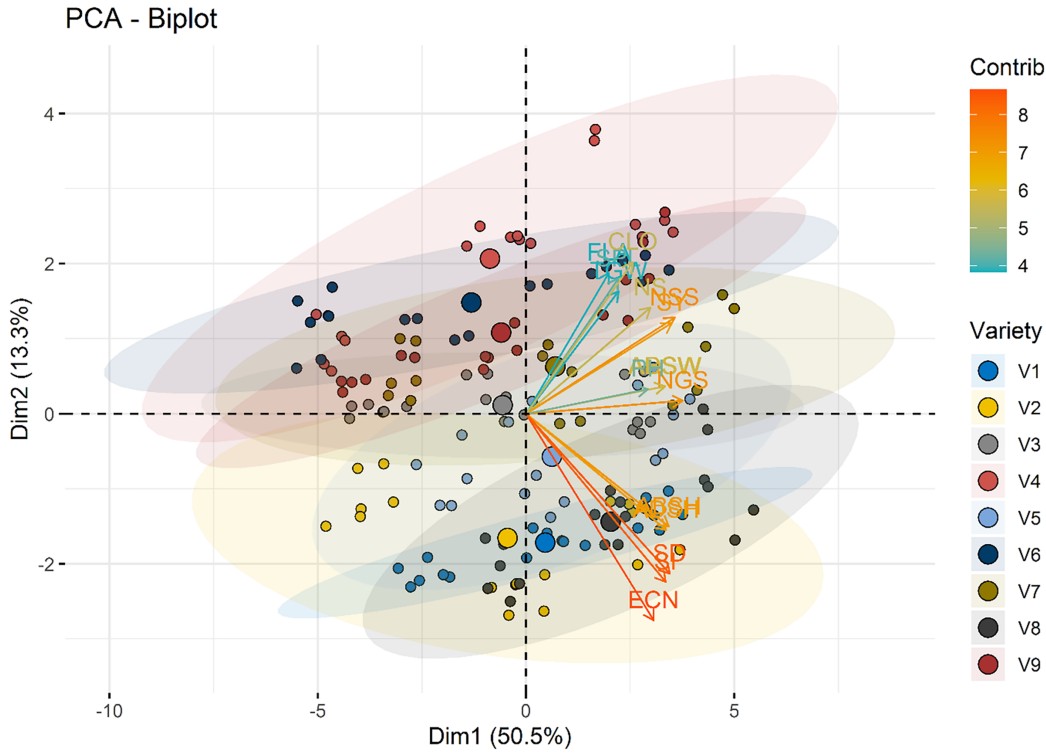

**Figure 9 PCA biplot illustrating the distribution of individuals across the first two dimensions (Dim1 and Dim2).** Variety groups (V1–V9) are color-coded, with ellipses representing 95% confidence intervals for each group. Variable loadings are depicted as arrows, highlighting their contributions to Dim1 and Dim2 and the group separation.

progressively with each increment of N from 0 to 150 kg ha$^{-1}$, further addition of N above 150 kg ha$^{-1}$ did not lead to additional increase, but rather decrease in seed yield ha$^{-1}$.

According to our results, chlorophyll content in the flag leaf varied among N doses and growth times. While chlorophyll content increased as N doses increased, chlorophyll values were found higher in early growth period than from late growth periods. N, which is found in the structure of chlorophylls that play a role in photosynthesis, is very important for chlorophylls (*Zangani et al., 2021*). Most of the N is found in the leaf in chloroplasts and mostly in ribulose bisphosphate carboxylase (*Makino et al., 1997*). Therefore, in case of N deficiency, there will be a decrease in rubisco activity and a decrease in photosynthesis. Photosynthesis plays a direct role in the growth, development and yield of plants. Therefore, changes in pigments that play a role in photosynthesis mean changes in the rate of photosynthesis in plants (*Sallam et al., 2019*). As a result, seed yield decreases (*Tóth et al., 2002*). The amount of chlorophyll varies between species and may also vary within a species. In addition, leaf structure is one of the important factors that determine the amount of chlorophyll (*Criado et al., 2007*). In our study the highest chlorophyll content was found of Bezostaja-1 and Nacibey wheat varieties (V4 and V7, respectively) both of years due to the highest flag leaf area. The excess chlorophyll content in plants stimulates growth and development while increasing yield. The study results confirm the data and draw attention to the role of N in yield increase (*Sharifi & Mohammadkhani,*

*2016*). In addition, the chlorophyll content of flag leaves in grasses can provide important clues about the growth and development status of plants (*Qian et al., 2021*; *Pandey et al., 2023*). Also, the amount of chlorophyll in plants varies depending on time during the vegetation period (*Čaňová, Ďurkovič & Hladká, 2008*) and it is closely related to plant photosynthetic capacity (*Zhang et al., 2022*), N is transferred from leaves to spikes, resulting in a decrease in spectral reflectance and chlorophyll content (*Zavoruev & Zavorueva, 2003*). *Shirazi et al. (2014)* reported that leaf area is an important indicator of crop growth. Biomass production in plants is closely related to leaf area (*Man, Yu & Shi, 2017*). *Li et al. (2023)* stated that the leaf area increases, the amount of biomass and seed yield also increases. While the leaf area was 25.28 cm$^2$ at the control dose, it increased to 31.80 cm$^2$ at the 100 kg ha$^{-1}$ dose. This showed an increase of 25.79% in the leaf area. The increase in biomass production will play an important role in improving the yield of wheat, and increased biomass production should be given priority to increase seed production of wheat (*Li et al., 2023*). In our study, an increase in leaf area was observed with increasing N doses, thus an increase in yield was observed.

Chlorophyll content in flag leaf starts to decrease with time after antesis, regardless of treatments and varieties, as plants undergo natural senescence during seed filling (*Kumar et al., 2022*). *Steer & Harrigan (1986)* reported an average decrease of 23% in chlorophyll content from antesis to physiological maturity (*Zhao et al., 2005*). In line with previous studies, our research findings indicate a decrease in chlorophyll content as the growth stages is delayed.

Decreases in chlorophyll content may mark the start of senescence (*Bassi, Menossi & Mattiello, 2018*). Indeed, in studies conducted in sugar cane, total chlorophyll content was found to be higher in young leaves and with increased N treatments (*Bassi, Menossi & Mattiello, 2018*). In our study, 1st growth stages (GS1) of wheat was found the highest chlorophyll content.

In addition, parameters such as the position of stomata on the leaf (being lower or upper) and stomata density, which play an active role in gas exchange parameters, are parameters that increase the functionality of the chlorophyll in photosynthesis (*Wasaya et al., 2021*). Therefore, the reason for the yield differences between varieties can be associated with changes in these parameters.

Stomata of flag leaf are an effective property in the compatibility of varieties to different environmental conditions, which can have an important role in gaining high performance. However, this effectiveness can be affected by environmental conditions and different growing seasons (*Anjum et al., 2011*; *Limochi & Eskandari, 2013*; *Caine et al., 2019*). The stomatal traits help plants maintain photosynthetic rates under water deficit (*Ding et al., 2018*) and high temperature (*Reynolds-Henne et al., 2010*; *Farooq et al., 2011*; *Sharma et al., 2015*; *Urban et al., 2017*; *Caine et al., 2023*). The open stomata maintain adequate gas exchange for photosynthesis, which is vital for plant metabolism (*Sayed, 2003*; *Reynolds-Henne et al., 2010*; *Sharma et al., 2015*; *Tricker et al., 2018*; *Zhao et al., 2020*). With increased photosynthesis, plant productivity increases. However, depending on the plant species, one or more of the same stress factors may have different effects on stomata

(*Merilo et al., 2014*; *Sharma et al., 2015*; *Sommer et al., 2023*) and, stomata density and size in cultivated wheat tend to be larger and sparser than in wild species (*Wang et al., 2022*).

The plants increase the number of stomata to minimize transpiration and water loss and closing stomata (*Yavas et al., 2024*). The researcher stated that planting date and varieties were found significant effects on all anatomical properties including stomata area, stomata diameter and stomata number of rice. Early planting (22 May) reduced the area and diameter of stomata, also grain yield. Grain yield showed maximum correlation with stomata area (0.405). Researchers suggested that this property can be used as an important factor in breeding programs for improving rice grain yield (*Limochi & Eskandari, 2013*).

In our study, all stomatal parameters increased with increasing N doses (Tables 5 and 6). and among the wheat varieties, the highest stomatal index and density were obtained from Harmankaya-99 wheat variety (V8) (Figs. 5E and 6E) in both years. It has been determined that the criteria considered in this study, together with N doses, show significant effects among varieties, and therefore yield increases are achieved at significant levels.

The effect of N fertilizer applications on stomatal conductance, flag leaf area and other physiological parameters of sorghum was found to be insignificant (*Bruns, 2016*). In another study, the N fertilizer increased, stomatal conductance of rapeseed increased as much as 21%, and it was not statistically significant for rapeseed (*Nasab et al., 2014*).

The flag leaf is the main component of the canopy in growth stages of wheat, and it determines the grain-filling rate and the yield (*Vicente et al., 2018*). Some studies have shown that the number and density of stomata significantly affect their ability to regulate photosynthesis and transpiration. The stomatal density of wheat flag leaves increased under limited water irrigation, and later irrigation resulted in greater stomatal density. Also, the stomatal length and width decreased, stomatal density of plant leaves increased and the stomata got smaller under stress conditions, but changes in the stomatal shape varied greatly (*Bertolino, Caine & Gray, 2019*; *Liu, Liao & Liu, 2021*).

In the present study, according to the principal component analysis, PCA biplot that visualizes the relationships between different varieties of a particular dataset. The x-axis represents the first principal component (Dim1), which accounts for 50.5% of the variance in the data, while the y-axis represents the second principal component (Dim2), which accounts for 13.3% of the variance. The data points, representing the different varieties, are colored and sized according to their contribution (Contrib) and variety (Variety), respectively. Varieties that are closer together on the plot are more similar to each other, while those that are farther apart are more distinct. The arrows indicate the direction and magnitude of the correlation between the principal components and the original variables. Varieties with longer arrows have a stronger influence on the principal components. This PCA biplot provides a useful overview of the underlying structure of the dataset, allowing the researcher to identify patterns, clusters, and relationships between the different varieties. The information gleaned from this analysis can inform further research, product development, or other desicion-making processes. *Koppensteiner et al. (2022)* stated that for experimental data at 0 g N m-2, PC1 explained 55.3% experimental data of all N fertilization rates, PC1 carried 51.5% and PC2 28.9% of variation summing up to 80.4%.

Similarly, *Zewdu, Mekonnen & Geleta (2024)* were the first four principal components accounted for 76.76% of the total observed phenotypic variation at the results of principal component analysis. *Tunç, Yılmaz & Yaman (2023)* stated that in their research on the olive varieties that the first five components consisted of 79.08% of the total variation in terms of stomatal morphology and chlorophyll content.

As researchers have stated that principal component analysis of yield and yield components in wheat generally show highly varying results which can be attributed to varying experimental and environmental conditions (*Ozukum et al., 2019*; *Nayana, Kumar & Chesneau, 2022*; *Bonfil et al., 2023*).

## CONCLUSIONS

This research has confirmed the hypothesis that nitrogen is an effective element on growth, development, productivity, chlorophyll content, abaxial stomata width, abaxial stomata length, adaxial stomata width, adaxial stomata length, epidermal cells, stomata cells, stomata density and stomata index in plants. In this 2-year study, the increase in N dose boosted the number of spikelets per plant, the number of seeds per spike and 1,000 seed weight, and accordingly, seed yield per hectare increased. In addition, as the leaf area index increased, so did the chlorophyll amount and stomata size and thus photosynthetic efficiency increased and accordingly, the amount of matter accumulated in the seed increased. It was determined that the best results in yield and other parameters examined were obtained from the 100 kg ha$^{-1}$ N dose. The hypothesis that nitrogen doses significantly boosted abaxial stomata width, abaxial stomata length, adaxial stomata width, adaxial stomata length, epidermal cell, stomata density, stomata index and stomata density was confirmed in this study. Significant increases were obtained with nitrogen doses in flag leaf area, which plays a role in photosynthetic efficiency. Yields were higher in varieties with denser stomata, while yields were lower in varieties with less dense stomata. More detailed studies should be conducted based on the results of this study to gain a deeper understanding of the relationships between stomatal densities and N fertilization, and thus yield increase.

### Funding
The author received no funding for this work.

### Competing Interests
The author declares that they have no competing interests.

### Author Contributions
- Fatih Oner conceived and designed the experiments, performed the experiments, analyzed the data, prepared figures and/or tables, authored or reviewed drafts of the article, and approved the final draft.

## Data Availability

The raw data are available in the Supplemental File.

## Supplemental Information

Supplemental information for this article can be found online at http://dx.doi.org/10.7717/peerj.18792#supplemental-information.

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
