# Peer review of "Effects of nitrogen doses on stomatal characteristics, chlorophyll content, and agronomic traits in wheat (Triticum aestivum L.)"

_PeerJ, doi:10.7717/peerj.18792_

## Round 0.1 · original submission · Major Revisions

In addition to reviewer comments, please also consider editorial comments. Some authors only make corrections requested by reviewers. This may result in your manuscript being reviewed longer or rejected.

-The English language must be reviewed by a native speaker and corrected grammatically. Some terminological errors should also be corrected.

-Reorganize your manuscript by taking into account the journal's guidlens from the link https://peerj.com/about/author-instructions/. Especially the citation format and references are not appropriate. Line numbers should be on the left.
-The abstract section is written very complicatedly. Please follow a structure that provides information about the purpose of the study, methods, results and conclusions.
-The introduction section detracts from the originality of the study. You should focus more on the effect of nitrogen on chlorophyll content and stomatal characteristics and emphasize the effect of these characteristics on yield. Also, do not ignore that nitrogen efficiency and yield estimates can be made from SPAD meter readings.
-There is an error in the statistical analysis. Please correct it. It is wrong to think of years as blocks. You can analyze years separately. It is also important to emphasize the differences between years. Also, it is a big deficiency not to measure at least grain N content. Mention the correlation analysis you conducted here. The study is quite weak in its current form.

Check your results. The conversion of yield values ​​to kg ha-1 is incorrect. Correct it. No one wants to grow a practice or variety that yields 51 kg per hectare. It should be 5129 kg or 5.13 tons.

The results and discussion sections need to be rewritten. Explain the results more briefly. It is pointless to repeat the numbers that we can already read from the tables. Instead, you can focus on percentage increases.
In the discussion section, instead of writing about the increase in yield of nitrogen doses that have been done thousands of times, focus on the importance of SPAD meter readings in determining the effectiveness of N doses. Also discuss which period readings can give us more information. Provide more information about stomatal characteristics and N applications and how the change in yield is affected. Adding these will increase the originality of your study. In this form, it is both unoriginal and incompatible with the title. Remember that how you explain it will increase the value of your study.

Check and correct abbreviations throughout the text. Do not write N in some places and nitrogen in others. Just write N to all of them.

Reviewer 1 ·

Basic reporting

The article is based on a fairly large amount of data, but I doubt its correctness due to the incomplete description in the Material and Methods section of all factors that could affect plant parameters, soil properties, measurement accuracy, etc. The statistical processing of the data is very questionable.

Experimental design

The research question is not clearly defined, but it is relevant and has practical significance. The study fills an existing gap in knowledge.

Validity of the findings

The results are not reliable enough due to incorrect statistical processing of data and fragmentary materials and research methods.

Additional comments

1. It would be better to remove the first sentences from the abstract: "Wheat is a grain plant that is very important for human nutrition. It is a very common plant that can be grown in various climate and soil conditions around the world. Since it is a plant widely produced and consumed by humans, a lot of work has been done to increase productivity and quality."
2. The second half of the abstract is overloaded with numbers that are not interesting and informative enough for the reader. The authors should rewrite the abstract, adapting it so that it is interesting and informative for readers to read this abstract in the Scopus or Web of Science databases.
3. At the end of the abstract, one sentence should be added about the prospects for research on this topic.
4. The hypothesis "This study was designed with the hypothesis that nitrogen doses have no effect on some physiological and agronomic traits of wheat" is poorly formulated.
5. The material and methods are not described in sufficient detail. I cannot understand what measurements and in what repetition the author made. With what accuracy were the measurements made? I have not found a detailed description of which parameters of the modeled system were under the author's control and which were not. My doubts are confirmed by the tables and figures in this article.
6. Table 1: are these one-off studies? Then the manuscript cannot be recommended for publication. How reliable are the differences between the years?
7. The statistical processing of Table 2 is unsatisfactory. The data should be presented in a form common to biomedical and agricultural sciences: mean value +- standard deviation. Samples that, according to the results of the Tukey test (or another sufficiently correct method of multiple comparison of samples), significantly differ from each other should be marked with different letters. In English, tenths are separated by a period, not a comma. The rounding of numbers should be correct. I recommend dividing Table 2 into several tables so that readers can easily work with this data.
8. The design of Table 3 does not allow it to be published. The same comments as for Table 2. The table is not "friendly" to the reader: you need to reread the entire "Material and Methods" section to understand the units of measurement, what the abbreviations mean, what the unit of measurement of the data is, what the method of comparing samples is, what the repetition is, and much more. Tables cannot be designed like this. Each table (figure) must be self-sufficient.
9. The same comments for Table 4. The letter next to the samples probably means something. Unfortunately, this is clear only to the author. The author must use different letters within a column (or row? - I don't know) to designate samples that are significantly different from each other based on the results of some fairly strict method of multiple sample comparison (for example, the Tukey test). In each cell of the table, you need to write the mean value +- standard deviation. 10. The same comments apply to Table 5 as to Tables 2 and 4.
11. In Figure 1 (ordinate axis), all figures should be rounded to whole numbers, not to tenths. The caption above the figure should be removed. What can the reader understand from these figures? Provide the necessary explanations on the figure itself or present the data in another form. This is primary data. Some statistical processing is required in science.
12. Figure 2 is actually a table. This data is not informative. This data should be processed using some correct method of multivariate statistics. For example, discriminant or factor analysis. Cluster analysis would be less successful. But in my opinion, providing a correlation matrix in a modern article is unacceptable.

Reviewer 2 ·

Basic reporting

There are some minor revisions required in the manuscript.

Please explain the gap in the literature in the Introduction section. This is mandatory for sowing the requirement of this study.

Please write the species name as italic (line 63-64).

There are enough numbers of literature. Tables and figures look appropriate for journal rules.

Experimental design

M&M section is well written and experimental design is appropriate. For split plot design maybe main and sub plot descriptions could be added. But citation for agronomic measurement methods could be added (line 94-95).

Validity of the findings

There is a duplicated title in the results section (lines between 251-277).

The correlation analysis looks not necessary. Because it is not present or discussed in the manuscript.

In the discussion part some of references should be corrected (there is no line number because it can not mention here).

Additional comments

The requested revisions are shown the reviewed file.

Annotated reviews are not available for download in order to protect the identity of reviewers who chose to remain anonymous.

Reviewer 3 ·

Basic reporting

English should be improved;
In example;
Abstract
*L13 “year taken as block” should be “year considered or used as block”
*L16-17 “1st growth stage chlorophyll content had highest rates (30.48)” should be “chlorophyll content was the highest (30.48) in the 1st growth stage of the plant”.
Introduction
*L34 aware wordiness “With the increase in population, the demand for grain production also increases.” It might be “The grain demand increases by the population”
*L67 “The objective of this study was to investigate determinate the effects of…” using both words the investigate and the determinate is meaningless. Please revise it.
Results
*Section 3.1. 5th sentence “When this increase is evaluated as a percentage, it is seen that it provides an increase of approximately 49.24%.” should be “The increment was approximately 49.24 %”.
* Section 3.1. the second paragraph “The plant height was significantly affected by N fertilizer, varieties, and their interaction wasn’t significantly affected.”
English should be edited totally.

* Literature and references are sufficient.
* Tables and figures are professionally presented

*The method should be more clear in the abstract. For example, experimental years should be given (i.e. 2023-2024).
*The given results are not clear in terms of variance source. For example, the author(s) stated “Flag leaf area ranged from 19.79 cm² to 41.86 cm² “. What is the source of variance? Are the results ranged depending on cultivars? or N applications? This should be clarified.
*L28 Triticum should be italic
*L37-38. Sentences are not fluent. The author (s) might consider combining two sentences here. However, these problems exist in several parts of the manuscript (i.e. section 3.1 grain yield, agronomic traits). So I suggest a detailed revision of the language and writing.
* Please use a space between numbers and the units.
* L41, 51, 54, 56… please delete the (-) inside the words.
* Font size is different in section 2.1 Experimental Design and Materials. Please change.
* L73-74 Please correct the sentence “The soil was neutral in response to (pH 5.78-6.62)” suggestion “The soil pH was neutral (5.78-6.62)”.
* While giving range use the lower value first (L74 phosphorus 0.20 – 0.19)
* Author(s) explained the variation of humidity in materials and methods (L85-88). Precipitation and temperature are common phenomena and therefore, I suggest precipitation and/or temperature variations should be explained instead of humidity as long as fungal diseases are not the subject.
*L92 “pod” is not a known term. It should be “plot”.
* Why flag leaf area not evaluated under the agronomic traits section? I think it should be moved to the agronomic traits section.

There are conflicts between hypotheses and results. They are explained in the next section.

Experimental design

* I think the originality of the research is low. The effect of nitrogen fertilization on wheat cultivars is a common subject and has been studied for almost half a century. I suggest that the author(s) might point out the differences clearly in terms of hypothesis and results. Because the results have low originality as it is. I do not believe that they will take an interest in the readers of the journal as long as the writing is not improved.
* There has been confusion with the hypothesis of the research. At the end of the Introduction, the author(s) stated “the hypothesis that nitrogen doses have no effect on some physiological and agronomic traits of wheat. The objective of this study was to investigate determinate the effects of N fertilization on some agronomic traits…”. As I understand, the author(s) tried to explain that relevant research results suggest that nitrogen is not effective on the agronomic characteristics of the wheat but, they tried to present vice versa in this study. If it is true, please make it clear and add this purpose to the abstract. However, plenty of results indicate the significant effects of nitrogen doses on agronomic traits of wheat. It is also discussed in the second sentence of the discussion. There is a contrast in the research question.

-Kayan, N., Kutlu, I., & Ayter, N. G. (2018). The influence of different tillage, crop rotations, and nitrogen levels on plant height, biological and grain yield in wheat. AgroLife Scientific Journal, 7(1).

-Ali, A., ……Rasul, F. (2012). How wheat responds to nitrogen in the field. A review. Crop and Environment, 3(1-2), 71-76.

-Galindo, F. S., ……Alves, C. J. (2019). Nitrogen fertilization efficiency and wheat grain yield affected by nitrogen doses and sources associated with Azospirillum brasilense. Acta Agriculturae Scandinavica, Section B—Soil & Plant Science, 69(7), 606-617.

*I did not understand why years were considered as block in the statistical analyses. I suggest year be considered as a random factor because the effect of the year should be separately investigated on varieties and fertilization.

Validity of the findings

* Please check grain yield results. I think there has been an error in conversion from decare to hectare.
*For height, spikelet number, grains per spike, and spike number results one decimal is enough (79.02 = 79.0 cm)

* In conclusion the author(s) stated that N use increased leaf area, chlorophyll content, stoma size, and photosynthesis. Therefore, accumulated substance amounts in grains. However, there is no evidence that these parameters could increase accumulated substance in grain. Therefore, this sentence should be revised correctly.

Additional comments

* This research should focus on stomatal characteristics and their effects on the grain yield. Otherwise, the study only remains a repeat of the many studies.
* Statistical procedure should be checked
* English should be revised totally.
* Hypothesis should be given clearly.

---

## Round 0.2 · Major Revisions

I would like to point out that the changes in accordance with the comments made, as stated by the reviewers, have not been fully implemented. It is not possible to accept your manuscript in this form. The effect of nitrogen doses on wheat yield is a subject that has been studied for almost a century and the way you present your data detracts from its originality. Please emphasise the effect of nitrogen on stomatal characters and chlorophyll content and its variation according to genotypes. Emphasising these will increase the originality and attractiveness of the study. You can also emphasise the yield-related characteristics. However, you should also relate yield traits with stomatal traits and chlorophyll measurements. The statistical analysis in the study is incorrect. As stated in the previous report, years should be taken as a factor, not as a block. Re-analyse and interpret your results.

In the introduction, describe the stomatal characteristics and chlorophyll changes due to nitrogen application. There is no need to read information that has been repeated hundreds of times. Hierarchical cluster analysis is irrelevant for this study. In fact, Reviewer 1 does not mean to do that instead of simple correlation analysis. You can make more accurate interpretations with principal component analysis. Especially emphasise the relationships arising from stomata and chlorophyll measurements. Direct the discussion accordingly. You should review more literature on stomata and chlorophyll measurements. In addition, it would be more appropriate to present your results graphically rather than in tables. Thus, you can overcome the complexity mentioned by Reviewer 1 with error bars on bar graphs and the letters you display above them. With the biplot graph obtained from PCA analysis, you can both interpret the correlation between traits and recommend the appropriate N dose for the appropriate genotype. You can take a look at the following article as an example for these corrections: Effects of different tillage, rotation systems and nitrogen levels on wheat yield and nitrogen use efficiency

If you do not fulfil these changes as requested, your manuscript will be rejected.

Reviewer 1 ·

Basic reporting

As a reviewer, I was not present during the laboratory stages of the study, but I can indirectly judge the authors' attention to measurements by the number of errors in the manuscript. If the authors are careless about the tables, then perhaps they were just as careless in measuring these characteristics?
I do not understand why the data in not all columns of each table in the article are statistically processed correctly. There is not always a +- standard deviation. There are not always letters next to the data. For example, in Table 4 there are columns without letters, and there are also columns without +- standard deviation, but with letters. This applies to all tables.
I do not understand the letter designations next to the numbers, when the authors use the symbols "-" or "+". The correctness and clarity of the data is very important for both the authors and the journal.
Rounding of figures with numerous errors, for example, "53.3+-1.21" in table 4 or "41±1.4" in table 9. The mean and standard deviation within one column should be rounded to the same digit capacity. There is a lot of such careless rounding of figures. The authors should check everything very carefully. The bottom line in the table with the same "<0.01" does not make any sense. This applies not only to table 4.
In the title of each table or in the note under the table, you should always indicate what is calculated in the table: the standard deviation or the standard error or 1.96 * standard deviation? Here you should also indicate the repetition of the measurement, for example, "n = 7".
There are a huge number of technical errors in the text of the article: a year is missing, there is no comma or period, a space is missing somewhere in many lines (for example, lines 88, 93, 363, 367).
Within the same brackets, sources should be arranged by year (e.g. lines 71, 78). Throughout the text of the article, % should be rounded to tenths: carelessness in figures indicates general carelessness in research (e.g. lines 47, 52, 201, 404). The text will not be understandable to readers, for example, "N doses, varieties and N dose x variety interactions..." (line 306) or " (682.8 no./mm2)" (line 308). The literature often lacks complete information, lacks periods or commas, or mixes up upper and lower case letters (e.g. lines 442, 446, 461, 470, 471).
I cannot recommend the article for publication in this form. Careful revision of this manuscript is required.

Experimental design

No comments.

Validity of the findings

No comments.

Additional comments

No comments.

Reviewer 2 ·

Basic reporting

There is no problem in terms of language and technical terms used in the manuscript. The number of literature is enough and related with the manuscript content. Figures and tables were given in appropriate form according to journal rules. Additionally, the requested revisions were made.

Experimental design

Experimental design is appropriate. Methods described with sufficient detail and it has enough info to replicate. When considering the previous manuscript file, it is shown that the requested revisions were made.

Validity of the findings

The results is discussed enough. There is a conclusion based on general results of the study. It is shown that the requested revisions were also made by author.

Additional comments

There is no additional comments.

Reviewer 3 ·

Basic reporting

* English is sufficient
* Literature and references are sufficient.
* Tables and figures are professionally presented

Experimental design

* I think the originality of the research is low. The effect of nitrogen fertilization on wheat cultivars is a common subject and has been studied for almost half a century. I suggest that the author(s) might point out the differences clearly in terms of hypothesis and results. Because the results have low originality and possibly, will not take an interest in the journal's readers.
* There has been confusion about the research hypothesis. At the end of the Introduction, the author(s) stated “the hypothesis that nitrogen doses do not affect some physiological and agronomic traits of wheat. The objective of this study was to investigate the effects of N fertilization on some agronomic traits…”. As I understand, the author(s) tried to explain that relevant research results suggest that nitrogen is not effective on the agronomic characteristics of the wheat but, they tried to present vice versa in this study. If it is true, please make it clear and add this purpose to the abstract. However, plenty of results indicate the significant effects of nitrogen doses on agronomic traits of wheat. It is also discussed in the second sentence of the discussion. There is a contrast in the research question.

*I did not understand why years were considered as block in the statistical analyses. I suggest year be considered as a random factor because the effect of the year should be separately investigated on varieties and fertilization.

Validity of the findings

* In conclusion the author(s) stated that N use increased leaf area, chlorophyll content, stoma size, and photosynthesis. Therefore, accumulated substance amounts in grains. However, there is no evidence that these parameters could increase accumulated substance in grain. Therefore, this sentence should be revised correctly.

Additional comments

* This research should focus on stomatal characteristics and their effects on the grain yield. Otherwise, the study only remains a repeat of the many studies.
* Statistical procedure is not proper

---

## Round 0.3 · Minor Revisions

Your efforts to improve your manuscript are appreciated, but it is still not without minor errors that you have missed. Please correct. Read your manuscript carefully to correct any typos.

Lines 37-38: The stomata density was decreased while N dose increased, however, the stomatal index increased. Revise the sentence.
Line 41: "affect of increasing N doses except for stomata index" This statement contradicts Line 38. Check.
Lines 93-94: Revise the sentence as "Stomatal conductance increases with N in some species such as wheat."
Line 113: Replace "replication" with "repetative."
Line 126: "some chemical and pysical SOIL characteristics of the experimental area" Revise the table title.
Line 144: Revise to "four-five" times. "Four" or "five" Which one.
Line 196: Does this sentence have a beginning? "In addition" is used to continue a sentence. It's like you started explaining something in the middle. Check.
Line 256: "respectively years" is not a correct usage. There are many errors like this. Check the English version.
Line 360-361: I don't remember explaining the abbreviations here before. Explain the abbreviations when they first appear or don't use any abbreviations at all. Also, after abbreviating the word "nitrogen" once throughout the text, write it as "N" in all other lines. Don't write it sometimes as a long and sometimes as an abbreviation. Ensure uniformity.
Line 441-445: Remove these lines and literatures since the study has no relation to drought tolerance.
Line 450: Correct it to "antesis". What is "antes"?
Line 470-472: delete.
It is difficult to understand why you gave examples from studies on drought stress in the discussion when you explained the effect of nitrogen on stomatal properties in the introduction. Please replace it with relevant literature.
The discussion on Principal Component Analysis is insufficient. Please explain how the features correlate with each other.

---

## Round 0.4 · accepted · Accept

The changes you have made are sufficient for your manuscript to be accepted, but when PeerJ staff sends you a galley proof, make the following changes as well. Since these are very minor changes, I don't see the need for another revision. However, be sure to add them during the galley proof stage.

Line 93: "bean (Phaseolus vulgaris L.)," delete this.
replace "repetetive" with "replication" throughout the text.
Add the sentence "More detailed studies should be conducted based on the results of this study to gain a deeper understanding of the relationships between stomatal densities and N fertilization, and thus yield increase." as the last sentence of the Conclusions section.
Also delete literature that is not included in the text from the references list.